# Tuna behaviour at anchored FADs inferred from Local Ecological Knowledge (LEK) of pole-and-line tuna fishers in the Maldives

Ahmed Riyaz Jauharee[1,2]*, Manuela Capello[2], Monique Simier[2], Fabien Forget[2], Mohamed Shiham Adam[3], Laurent Dagorn[2]

1 Ministry of Fisheries, Marine Resources and Agriculture, Maldives Marine Research Institute, Malé, Maldives, 2 MARBEC, University of Montpellier, CNRS, Ifremer, IRD, Sète, France, 3 International Pole and Line Foundation, Reading, United Kingdom

* arjauharee@gmail.com

**Data Availability Statement:** All relevant data are within the paper and its Supporting Information files.

## Abstract

The Maldives tuna fishery landings in 2018 were 148, 000 t and accounted for nearly a quarter of the global pole-and-line tuna catch. This fishery partially relies on a network of 55 anchored fish aggregating devices (AFADs) deployed around the archipelago. About one-third of the total pole-and-line tuna catch is harvested at AFADs. Although the AFAD fishery has existed for 35 years, knowledge on the behaviour of tuna in the AFAD array is still limited, precluding the development of science-based fishery management. In this study, local ecological knowledge (LEK) of fishers was used to improve our understanding of tuna behaviour, through personal interviews of 54 pole-and-line fishers from different parts of the archipelago. Interview results suggest that during the northeast monsoon tuna are more abundant on the eastern side of the Maldives, while during the southwest monsoon they are more abundant on the western side of the Maldives. Most fishers believed that tuna tend to stay at the AFADs for 3 to 6 days and remain within 2 miles from the AFADs when they are associated. Fishers believe that strong currents is the main factor for tuna departure from AFADs, though high sea surface temperatures and stormy conditions were also thought to contribute to departures. Moderate currents are believed to be a favourable condition to form aggregations at the AFADs while other factors such as suitable temperature, prey and attractants enhance this aggregation. Fishers also believe that there are multiple schools segregated according to size and species at AFADs and that catchability is higher at dawn and in the late afternoon when the tuna occur shallower in the water column. This study is an important step towards engaging the Maldivian tuna fishers into a science-based fishery management.

## Introduction

Understanding fish behaviour is a key element of scientific expertise to assist in stock assessment and fisheries management [1]. Behaviour structures the spatio-temporal distribution of fish in three dimensions and defines their accessibility and vulnerability to fishing gear.

**Funding:** This study was fully funded by the International Pole & Line Foundation (IPNLF).

**Competing interests:** The authors have declared that no competing interests exist.

Collecting information on the behaviour of pelagic fish is quite challenging because of the difficulties in accessing these animals in their natural environment. The vast majority of the scientific knowledge on the behaviour of pelagic fish has been acquired through acoustic devices (sonars), visual devices (aerial surveys or cameras) and electronic tagging. These techniques, however, are expensive and require advanced scientific expertise. These constraints can limit their use, particularly in developing countries.

It is widely known that fishers spend a lot of time observing, understanding and accumulating information on fish behaviour [2–4]. Their fishing efficiency partly depends on this knowledge. Local Ecological Knowledge (LEK) or Fishers Ecological Knowledge (FEK) has therefore been used to make this knowledge available to science, as an alternative source of information [5]. For instance, LEK has been used to study migration [6], spatial distribution [7], habitat use [8], meso-scale behaviour [9], and fine-scale behaviour of fishes [10].

Several pelagic species, including tropical tuna, are naturally attracted to floating objects such as drifting logs or marine debris [11]. Although several hypotheses have been postulated to explain the associative behaviour of tuna [12], including the meeting point hypothesis [13, 14] and the indicator-log hypothesis [15], we still do not know why tuna associate with these objects. Naturally, this has not prevented fishers from taking advantage of this particular behaviour to help them find and catch tuna. The Roman author Oppian (200 AD) first reported catches of pelagic fish around floating objects in the Mediterranean sea, while aggregating devices were used in Japan in the 17th century [16]. The use of floating objects by fishers developed considerably in the 20th and 21st centuries with the expansion of fisheries targeting tropical tuna and the use of Fish Aggregating Devices (FADs). FADs are man-made objects built and deployed for the purpose of fishing. FADs contribute to increase the catchability of tuna and other pelagic species [17]. There are two types of FADs: drifting FADs (DFADs), usually equipped with electronic buoys to remotely locate them and sometimes send information on the quantity of associated fish [18], exploited offshore by industrial purse seiners, and anchored FADs (AFADs), primarily used near the coasts by small-scale fisheries [19].

Because FADs play such a major role in the efficiency of tuna fisheries, they must be properly managed. Regional Fisheries Management Organisations (RFMOs) such as the Indian Ocean Tuna Commission (IOTC) have therefore prioritized the establishment of FAD management plans [20] to ensure the sustainability of fisheries. Just like the technical and socio-economic characteristics of a fishery, or the functioning of the ecosystem, the behaviour of fish is a scientific knowledge necessary to establish coherent FAD management plans. The behaviour of fish at FADs has been investigated through acoustic devices (e.g. [21, 22]) and electronic tagging (e.g. [23–26]). LEK was also used to inform on the behaviour of tuna at DFADs [27] or at AFADs in the Philippines [28].

The Maldivian tuna fishery has traditionally been important. The tuna fishery has increased production with catches increasing from 30,000 tons in 1980 to 148,000 tons in 2018 (Ministry of Fisheries, Marine Resources and Agriculture / MoFMRA, 2019). Although this fishery uses AFADs since the 1980s, only one study investigated the behaviour of tuna at AFADs in the Maldives, through acoustic tagging [29].

The objective of this study is to use LEK to improve our knowledge of tuna behaviour at AFADs in the Maldives, at the scale of the FAD array (including seasonal variations) and at the scale of individual AFADs.

## The Maldivian tuna fishery and its management

The Maldives tuna fishery has existed for over a millennium [30]. Unlike many island nations, Maldivians depend more on pelagic fish resources than on coastal fish resources. Until tourism

started to expand in the 1980s, coastal fish were never targeted on a commercial scale while tuna was being harvested for both local consumption and export. Tuna represents 98% of the total marine catches in the Maldives (MoFMRA, 2019).

Throughout the Maldives only hook and line (pole-and-line, handline, trolling and long-line) fishing is practiced. About 75% of the tuna catch, mainly skipjack (*Katsuwonus pelamis*) and small yellowfin tuna (*Thunnus albacares*) (fork length <65cm), is caught using pole-and-line [31] while the rest is caught by handline and trolling. Although longline fishing for tuna occurred in the Maldives, it has been suspended since July 2019. Other gears such as purse seine, large gill nets or trawl nets were never used for fishing in the Maldives and are forbidden to ensure the sustainability of the stocks. At present there are no foreign fishing vessels or fleets operating in the Maldives EEZ. About half the total tuna catch landed in the Maldives (148,000 tons in 2018) is consumed locally.

The traditional livebait pole-and-line fishing technique has not changed much over the years, although fishers moved from small wooden sail boats (8 to 12 m in length) to large (25 to 30 m in length) mechanized vessels [32]. Within the pole-and-line tuna fishery there are two distinct categories: the artisanal fleet and the commercial fleet. The artisanal fleet comprises of small vessels (<20 m in length) with 8 to 12 crew members conducting day trips and usually selling their catches in the neighboring islands mainly for local consumption. These small vessels do not hold ice and cannot keep the fish fresh for more than a day. The commercial fleet comprises of large vessels (about 25 m to 30 m in length) with 20 to 30 crew members. These large vessels are designed to cope with rough sea conditions and can stay out at sea for several days. They have insulated holds and carry ice to keep the fish fresh. Most of these commercial pole-and-line vessels operate in the south of the Maldives but they can travel north to pursue better fishing grounds. There were 785 licensed local commercial tuna fishing vessels operating in the Maldives in 2018 (MoFMRA, 2019). These vessels are owned by families or individuals living in the Maldives. There are no company owned fleets in the Maldives. More than 17,500 active fishers (MoFMRA, 2019) work on these fishing vessels which operate within the Maldives EEZ.

In the Maldives skipjack and small yellowfin tuna are caught by targeting (i) free swimming schools (45.4%), (ii) logs or other drifting objects (11.3%), (which includes drifting fish aggregating devices (DFADs) deployed by purse seiners that pass through the Maldives EEZ), (iii) seamount associated schools (10.4%), and (iv) anchored fish aggregating devices (AFADs) (32.8%) [33]. Studies showed that AFADs related catches contribute to nearly one third of all the tuna caught by pole-and-line in the Maldives [33]. Traditionally, Maldivian tuna fishers have fished at logs or drifting objects associated schools for centuries and they refer to these objects that attract tuna as '*oivaali*' (a local name given to drifting objects). Thus, in the Maldives AFADs are called '*oivaali kandhufathi*'.

The AFAD fishery in the Maldives began in 1981 [34], with experimental FADs deployed by the Ministry of Fisheries and Agriculture. Studies conducted by Anderson and Waheed suggested that the expansion of the AFADs network would help further increase tuna catches in the Maldives [35]. With the success of this fishery, the number of AFADs deployed increased from an initial 10 (in late 1980s) to 55 AFADs in 2019 (Fig 1). Currently the AFADs in the Maldives are deployed on average about 20 km from shore, at depths between 1000 to 2800 m (MoFMRA, 2019). All AFADs are constructed, deployed, maintained and managed by the government thus all pole-and-line vessels across the Maldives (both artisanal and commercial) have equal access to all the AFADs. During a single fishing trip one vessel may fish at several AFADs and there can be as many as 30 vessels fishing at one AFAD. Only pole-and-line fishing is permitted within a 3-mile radius of the AFADs. Outside the 3-mile radius of the AFADs, all permitted fishing gears in the Maldives can be used for fishing.

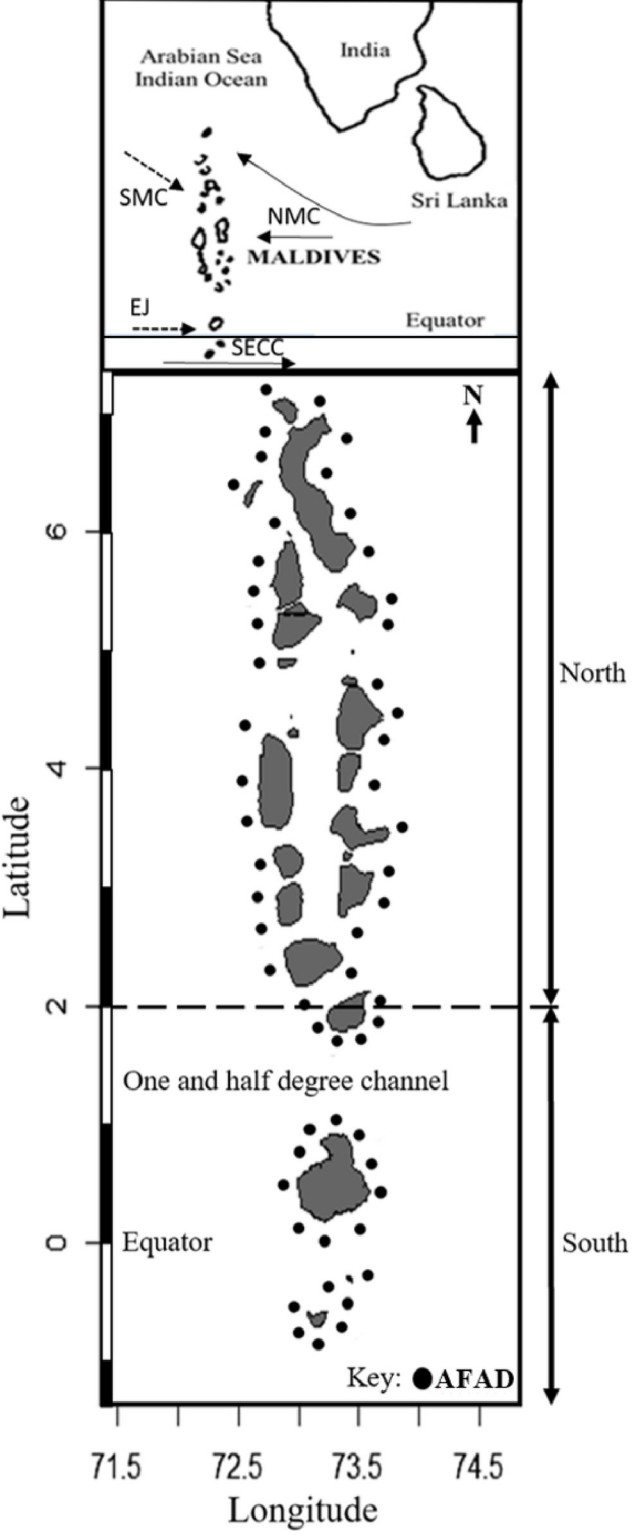

**Fig 1. The study area, AFAD network outside Maldives atolls and the direction of monsoon related currents.**
(Dotted arrows indicate southwest monsoon currents and the continuous arrows indication northeast monsoon currents. NMC–northeast monsoon current, SECC–south equatorial counter current, SMC–southwest monsoon current, EJ–equatorial jet).

A particularity of the AFAD array in the Maldives is the low density of these devices, (i.e. the large distances between neighboring FADs) [36]. While other AFAD arrays in the world are characterized by short distances between FADs, (e.g. 2–14 km in Mauritius [26] or 7–31 km around Oahu, Hawaii [24]) in the Maldives the distances between neighboring AFADs range from 25–48 km. FAD density (or inter-FAD distances) is typically a parameter that can be managed by governments or fishers. It can have impacts on fish behaviour and consequently on catches. It is therefore important to investigate the effects of this parameter on tuna behaviour through the comparison of fish behaviour in AFAD arrays differing by their inter-FAD distances [36].

## Materials and methods

### Study site

The Maldives extends from about 7˚N to 0.5˚S stretching 822 km (Fig 1) and is 130 km at its widest. It is subjected to the seasonal monsoons [37, 38]–northeast monsoon is from December to March and the southwest monsoon is from May to October. From almost the northern tip of the Maldives up to about 2.5˚N, the Maldives atolls are arranged in a double chain and below 2.5˚N the atolls form a single chain and are separated with wide deep channels through which migratory fish such as tuna travel. In the south of the Maldives, the effect of the monsoon related currents is diminished and influenced by the equatorial systems [37, 38]. More than two thirds of the yearly pole-and-line catch are landed by vessels operated in the south of the Maldives (Average pole-and-line catch for last 5 years: North (7˚ to 2˚N) 16,000 t (SD ±1000 t) and South (2˚N to 1˚S) 49,500 t (SD±6000 t) (MoFMRA, 2019)).

### Fishers interviews

In the Maldivian tuna fishery, the captains or fishing masters (Locally called—*Keyolhu*) start their fishing career as a crew member on a pole-and-line vessel and work their way up to become a captain. They are responsible for making the decisions related to the fishing strategy–based on the information obtained from the surrounding environment by visual observations using binoculars and bird radar. In addition, the fishing strategy is also influenced by prior knowledge on the oceanographic conditions, weather and recent fishing activities in the area. Based on this information, the captain sets the course and steers the vessel. The captain is also responsible for taking appropriate decisions when approaching tuna schools and manoeuvring the vessel during fishing events. The deputy captain works closely with the captain and during his absence, makes all the necessary decisions.

To ensure the validity of LEK and the quality of information obtained, it is important to select participants who have the appropriate knowledge for the interviews [10]. The fishers for the interviews were selected based on their fishing experience and area of fishing. Hence captains (n = 36), deputy captains (n = 9) and crew members (n = 9) from 36 vessels, who had a minimum of 8 years of experience on a licensed commercial pole-and-line vessel in the Maldives were selected. For this study, the Maldives archipelago was divided into north and south area at 2.0˚N (Fig 1) based on the physiographic differences of the study site [37, 38] and fishing practices of the local tuna fishers. To limit potential geographical biases on the response of fishers from the north and the south of the archipelago, 34 fishers from the south and 20 fishers from the north were selected.

A questionnaire was developed to obtain fishers perceptions on tuna aggregations at AFADs. Questions were addressed through personal interviews conducted in the local language (Dhivehi) at commercial fish landing sites and fishers home ports (local islands) where their vessel is based. The questionnaire was not filled in the presence of the fishers since most fishers are reluctant to express freely when the interview becomes too formal. Interviews lasted

about 30 minutes. All 54 fishers who contributed to the study provided verbal consent. No written consent was obtained since fishers were reluctant to sign any such documents. The study was approved by Marine Research Centre of the Ministry of Fisheries Marine Resources and Agriculture, Maldives.

To ensure consistency in the interpretations of the responses, all the interviews were conducted by the same individual (who had worked with tuna fishers for the last 10 years) in a friendly atmosphere and in an informal manner during 2017 and 2018. All fishers contributed enthusiastically to the interviews. For each question, the responses of the fishers were aggregated and converted into percentages. The questionnaire consisted of 11 questions and is included in S1 Appendix.

## Logbook data

All licensed pole-and-line fishing vessels report the mid-day position of the vessels during fishing operations and their daily catch in the logbook. The mid-day positions of the fishing vessels reported for the two monsoons, northeast monsoon (December to March) and southwest monsoon (May to October) were aggregated and used for calculating the percentage number of trips made during each monsoon season on the east and west side of the Maldives. All reported position between 70˚E to 73˚E were recorded as West and positions between 73.5˚E to 76.5˚E were recorded as East of the Maldives. Mid-day positions reported in the Central region 73˚E to 73.5˚E were not considered. April and November were considered as inter-monsoon periods and not included in the analysis. Logbook data of 2016 and 2017 were only used to assess the seasonal variation in fishing grounds reported by fishers.

## Statistical analyses

Fishers responses on the months where tuna abundance is higher (question 1 of the questionnaire–see S1 Appendix) concerned five school types (AFAD, Log schools, DFAD, Free school, Seamount) and the two main target tuna species skipjack (*Katsuwonus pelamis*) and yellowfin tuna (*Thunnus albacares*). The responses were coded "1" for "yes" and "0" for "no" and were analysed by developing a data table with 12 columns (the months) and 540 rows (the answers of the 54 fishers for the 5 school types and the 2 species). Multidimensional data from LEK questionnaires are often analyzed using multivariate approaches such as nDMS and PERMA-NOVA [39] Multiple Correspondence Analysis [8], and Principal Component Analysis [40]. Here, the table was subjected to a Principal Component Analysis on covariance matrix (centered PCA) in order to obtain an overview of the seasonality of tuna abundance, in relation with the species and the type of school but also with the origin and the position of the fishers. Four between-groups PCAs, a particular case of PCA with respect to instrumental variables [41] in which there is only a single factor as explanatory variable, were then performed to compute the ratio of variability explained by (i) the position of fishers (captains, deputy captains and crew members), (ii) the origin of fishers (north and south), (iii) the different types of schools and (iv) the species (skipjack and yellowfin). Monte-Carlo permutation tests [42] with 1000 random permutations were finally conducted to assess the significance level of the observed ratios in between-groups analyses. Chi-2 tests were used to compare the number of visits reported in the logbook between East and West of the Maldives. All statistical analyses were performed using R software [43] with the ade4 package [44].

## Results

In general fishers were very cooperative in sharing their knowledge on the AFAD fishery and the behaviour of tuna around the AFADs. Several of them were keen to provide additional

information that were not addressed in the questionnaire. All 54 fishers who took part in the interviews had no knowledge of the scientific publications related to tuna behaviour around FADs. The responses provided by the fishers at the scale of the AFAD array (questions 1 to 5, See S1 Appendix) and at the scale of individual AFADs (6 to 11, See S1 Appendix) are presented in Tables 1 and 2, respectively. Maldivian tuna fishers catch tuna from floating objects associated schools (AFADs, DFADs and logs), seamounts and free-swimming schools (Fig 2). AFADs are frequently used throughout the year by tuna fishers.

## Tuna behaviour at the scale of the AFAD array

In the interviews, fishers globally agreed that there is a seasonal variation in abundance of tuna in the Maldives around the AFADs (Table 1). All fishers observed that during the northeast monsoon there are more tuna on the east side and during the southwest monsoon there are more tuna on the west side of the Maldives. The mid-day positions of pole-and-line fishing vessels obtained from logbooks during the northeast and southwest monsoons (Fig 3) showed a similar trend, with fishers tending to fish more on the east (45%—fishing events) than on the west (32%—fishing events) of the Maldives during the northeast monsoon (Contingency test, Chi-square = 95.368, df = 1, p-value<0.0001) and on the west (42%—fishing events) than on the east (39%—fishing events) of the Maldives during the southwest monsoon (Contingency test, Chi-square = 6.743, df = 1, p-value = 0.009). The remaining vessels reported the central region as their mid-day position. When specifically asked about the tuna abundance throughout the year, it was highlighted that most skipjack and yellowfin were found around AFADs during the northeast monsoon.

The first two axes of the PCA (Fig 4) summarized respectively 20% and 14% of the information contained in the responses of the fishers about the seasonal variability of tuna abundance. The first axis opposed positive answers (all month projected on the left side) to negative answers (no month on the right side) (Fig 4A). As responses relative to AFADs were located mainly on the left side (Fig 4D), AFADs were considered by the fishers as globally attractive all-around the year. On the right side of axis 1 were the Log schools, Free schools and Seamounts, considered as globally less attracting fish. The second axis identified two groups of months: September and November to February on the upper side and the other months on the lower side (Fig 4A). The grouping of responses by school type (Fig 4D) allowed to conclude that, according to the fishers, more tunas were present around DFADs during the northeast monsoon, while more tunas were present around AFADs towards the beginning and end of the southwest monsoon.

Between-groups analyses, and the associated Monte-Carlo permutation tests confirmed that the abundance of fish over the months varied significantly according to the different types

**Table 1. Fishers' response to seasonal variation and association behaviour of tuna at AFADs, at the scale of the AFAD array.**

| Questions | Percentage (%) | |
|---|---|---|
| There is seasonal variation in abundance/size of tuna around AFADs on the east and west of Maldives | Yes 94.4 | No 5.6 |
| There are more fish at AFADs on the east side of the Maldives during northeast monsoon | Yes 100 | No 0.0 |
| There are more fish at AFADs on the west side of the Maldives during southwest monsoon | Yes 100 | No 0.0 |
| When fish are present in the AFAD array–two adjacent AFADs do not have same amount of tuna | Yes 88.9 | No 11.1 |
| There are AFADs that always attract less tuna | Yes 79.6 | No 20.4 |
| There are AFADs that always attract more tuna | Yes 87.0 | No 13.0 |

**Table 2. Fishers' responses related to formation of aggregation at AFADs, at the scale of individual AFADs.**

| Question | Percentage (%) | |
|---|---|---|
| There are multiple schools at AFADs and are segregated according to fish size and species | Yes 64.8 | No 35.2 |
| Time of the day influence the behaviour of tuna at the AFADs | Yes | No |
| Horizontal distance from AFAD | 11.1 | 88.9 |
| Vertical distance from AFAD | 100 | 0.0 |
| Catchability | 100 | 0.0 |
| Number of days that tuna aggregation is retained at an AFAD | | |
| Less than 3 days | 29.6 | |
| 3 to 6 days | 40.7 | |
| 7 to 10 days | 20.4 | |
| More than 10 days | 9.3 | |
| Distance fishers consider that the tuna is attracted to the AFADs | | |
| 0 to 2 miles | 74.1 | |
| 0 to 5 miles | 18.5 | |
| > 5 miles | 7.4 | |
| Reasons for tuna to aggregate at AFADs? | Yes | No |
| Moderate current (1 to 4 knots) | 83.3 | 16.7 |
| Suitable temperature | 37.0 | 63.0 |
| Less turbid | 20.4 | 79.6 |
| Presence of prey | 46.3 | 53.7 |
| Presence of sharks | 18.5 | 81.5 |
| Attractants present | 48.1 | 51.9 |
| Sea state (average) | 29.6 | 70.4 |
| Reasons for tuna to leave the AFADs? | Yes | No |
| Strong current (>4 knots) | 85.2 | 14.8 |
| High sea surface temperature | 40.7 | 59.3 |
| Turbidity (very high) | 13.0 | 87.0 |
| Absence of prey | 40.7 | 59.3 |
| Presence of predators/mammals | 29.6 | 70.4 |
| Attractants absent | 55.6 | 44.4 |
| Storms / very rough sea condition | 37.0 | 63.0 |
| Large size of aggregations | 24.1 | 75.9 |

of schools (18.1% of explained variability–p = 0.001). However, there was no significant difference (p = 0.133) between skipjack and yellowfin tuna. The responses of the fishers according to their position (captain, deputy captain, and crew) did not differ (p = 0.515) while responses according to their origin (from the south and north of the Maldives) differed significantly (p = 0.002) but only explained 0.6% of the variability of the data table.

Almost 90% of the fishers said that when tuna are present in the AFADs array, two adjacent AFADs never had equal amounts of tuna (Table 1). Almost 80% of the fishers consider that there are AFADs that always attract less tuna than others, while 87% of the fishers consider that there are AFADs that always attract more tuna.

## Tuna behaviour at individual AFADs

Most of the fishers (64.8%) believe that there are multiple schools of tuna at the AFADs while others (35.2%) think that there is one large mixed school (Table 2). All fishers observed that tuna move closer to the surface during sunrise and later afternoon, increasing their catchability

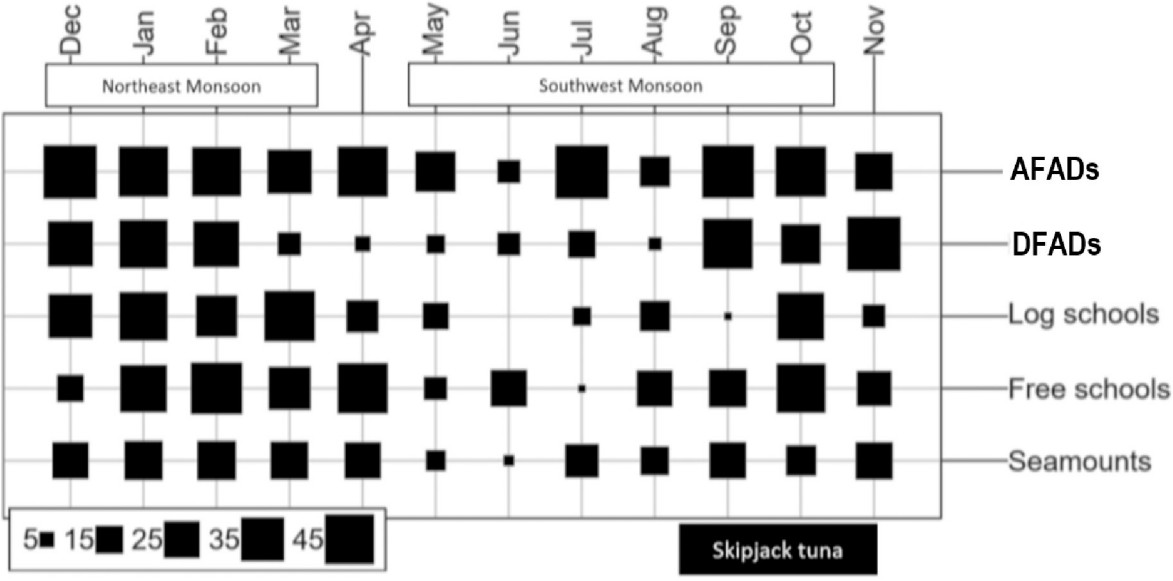

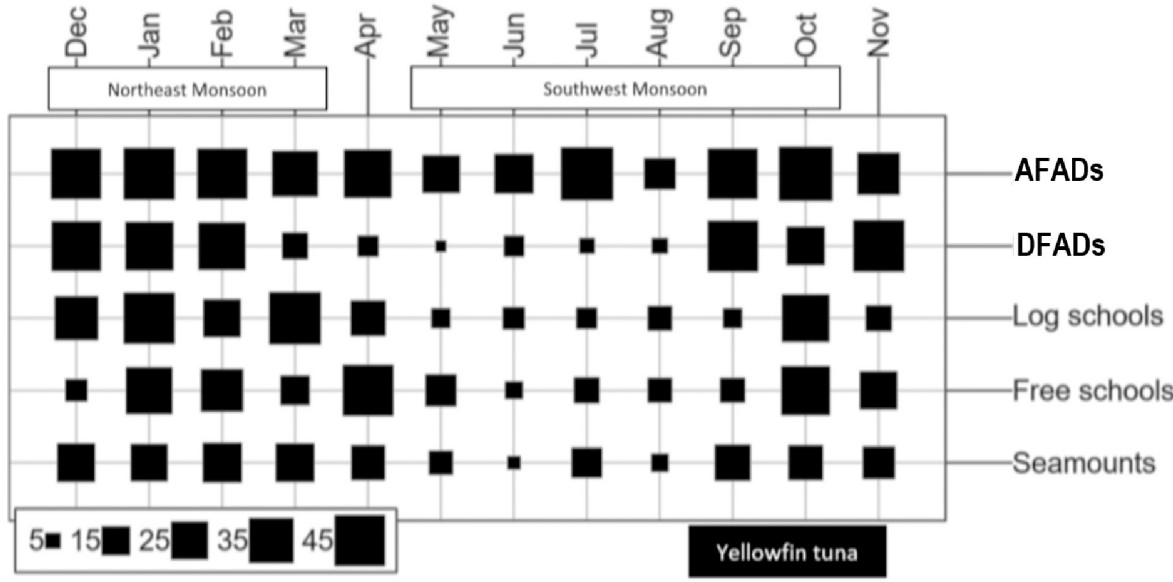

**Fig 2. Fishers' response on the seasonal variation in abundance of two tuna species at five types of schools in the Maldives.** April and November are inter-monsoon periods.

at AFADs. Only a few fishers (11%) thought that there is a variation in the horizontal distance of tuna from AFADs over the day. Nearly half (40.7%) of the fishers interviewed thought that tuna stayed at the AFADs for 3 to 6 days while nearly one third (29.6%) believed that they stay less than 3 days. Few fishers (9.3%) suggested that aggregations can last for more than 10 days (but they specified that this occurs at very few AFADs during very good fishing periods). Most of the fishers (74%) suggested that tuna could be attracted to AFADs from up to 2 miles. A few

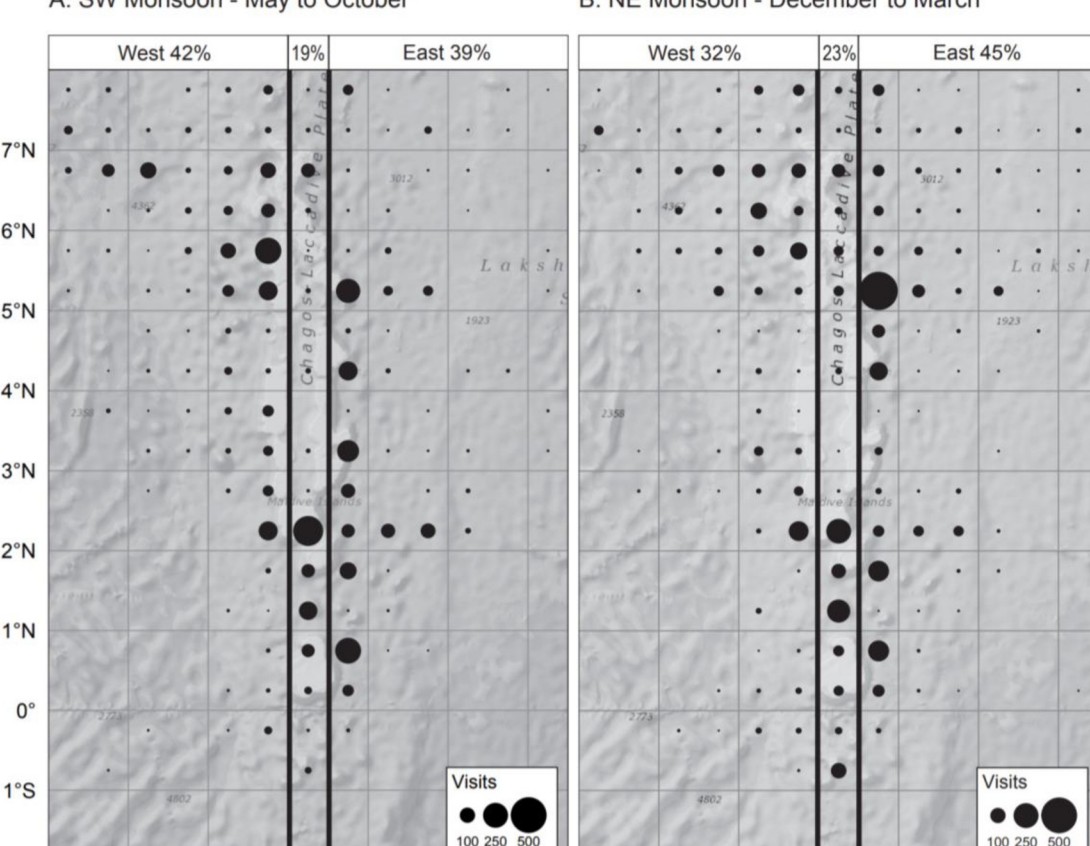

**Fig 3. Mid-day position of pole-line-fishing vessels from logbook data for the SW and the NE monsoon periods for the west (70˚E to 73˚E), central (73˚E to 73.5˚E) and east (73.5˚E to 76.5˚E) regions.**

(18%) suggested that it could extend to 5 miles. Some fishers (10%) suggested that this range could be area and season specific too.

Most of the fishers (83%) believed that current was the most important factor that drives tuna aggregations (Table 2). Moderate currents help to aggregate tuna at the AFADs. Almost half of the fishers (48%) thought that attractants (additional structures such as floats and ropes attached to the main buoy of the FAD) on the AFADs also contribute to the formation of aggregations while 46% fishers agreed that presence of prey (food for tuna) is also an important factor for the aggregations to form. About one-third (37%) of the fishers identified suitable sea surface temperature and average sea condition as other important factors. Some fishers believed that less turbid waters (20%) and sharks (18%) also play a role in the formation of aggregations.

Strong currents were identified by fishers (85%) as the most important factor that lead to the departure of tuna from AFADs (Table 2). Nearly half of the fishers (55.6%) thought that loss of attractants on the FADs could also result in the departure of tuna. High sea surface temperature and absence of prey were also identified as factors by 40.7% of fishers. Stormy conditions such as very rough seas (37%) and predators such as dolphins (29.6%) were believed to cause tuna to leave AFADs. Few fishers (13%) thought that high turbidity could also contributed to tuna departure.

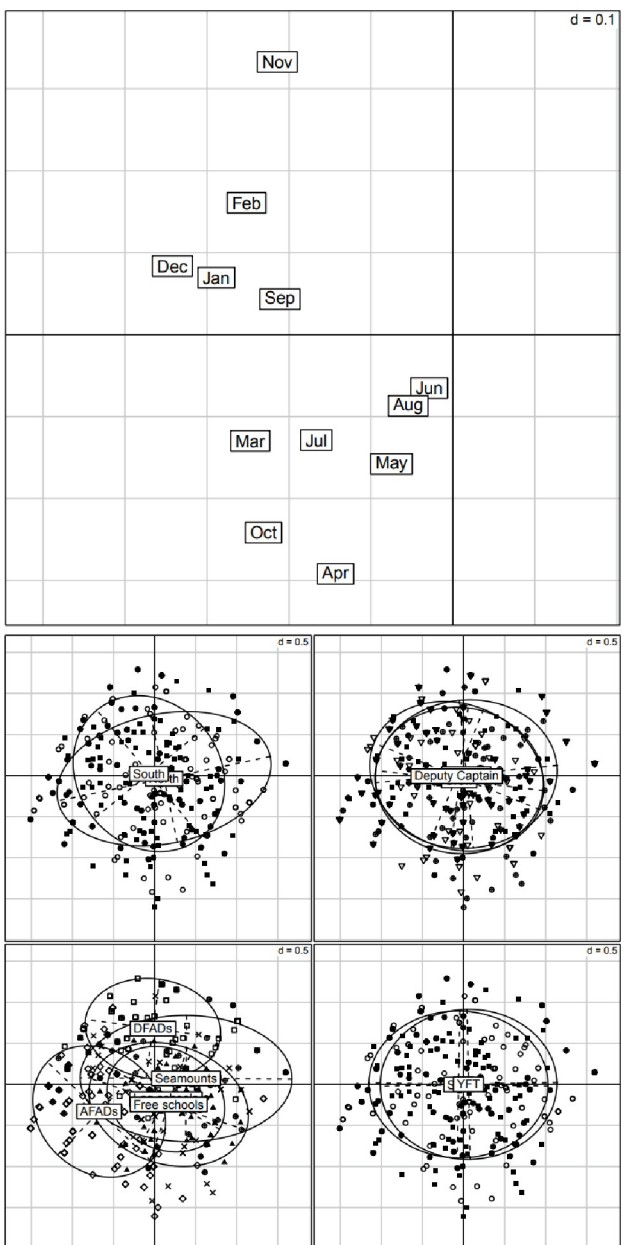

**Fig 4. Results of the PCA on the monthly variability of tuna abundance.** On the first two axes, representing respectively 20% and 14% of the total information, (A) projection of the 12 months and of the 540 rows grouped by: (B) Origin of the fisher—North (black squares), South (empty circles); (C) Position of the fisher—Captain (black square), Crew (empty triangles), Deputy Captain (crosses); (D) Type of school—AFADs (empty diamonds), DFADs (empty circles), Log Schools (black triangles), Free schools (black squares), Seamounts (crosses) and (E): species SKJ (black squares), YFT (empty circles). The value of d in the top-right corner gives the scale of the grid.

## Discussion

This study provides an insight into the spatio-temporal distribution and behaviour of tuna in the Maldives through the knowledge of tuna behaviour that fishers have observed during several years at sea. In the Maldives tuna fishers do not use sonars or echosounders for locating or observing tuna schools during their pole-and-line operation hence fisher's perceptions of tuna

behaviour were completely based on what they had observed at sea during their fishing events. These experienced fishers, over the years, have accumulated vast amounts of knowledge on various aspects of tuna behaviour.

The seasonal variation in tuna abundance in the Maldives observed by the fishers (increase of abundance on the east of the Maldives during northeast monsoon and more tuna found on the west of the Maldives during southwest monsoon) and confirmed through logbook data, is similar to what was reported by Adam et al. [45]. In general, there are high catches observed during the northeast monsoon (lasts for 4 months) than in southwest monsoon (lasts for 6 months). Over the last five years (2014 to 2018) MoFMRA catch statistics showed that the average catches during northeast monsoon (during a 4 months period) was 50,000±2000 t while in the southwest monsoon (during a 6 months period) it was at 54,000±3000 t. This could be due to severe weather conditions experienced during the southwest monsoon, making it difficult to fish. Fishers encounter more natural floating objects (log) associated schools during the northeast monsoon (Fig 2) and believe that tuna associated to floating objects and free-swimming schools of tuna move towards the Maldives with the monsoon currents. Similar observations were made by Adam et al. [45], with a pattern that could correspond to tuna entering the Maldives from the north (on the east side) at the beginning of the northeast monsoon and tuna entering from the south (on the west side) towards the beginning of the southwest monsoon.

Almost all the fishers agreed that they have never observed equal amounts of tuna at two adjacent AFADs but a few said that, when tuna was very abundant around the atolls they had, on few occasions, seen similar aggregations at two adjacent AFADs. The origin of this question relates to the fact that for social species (e.g. tuna), the competition of two attracting devices could lead to one of the devices aggregating most of the local population [46]. Robert et al. [25] designed an experiment with AFADs in the Seychelles to test this hypothesis on tuna and showed that tuna generally aggregate to one of the two close FADs. In their experiment, however, FADs were only 5 km apart. In the Maldives, two adjacent FADs are always more than 35km. According to fishers, although FADs are quite far from each other, the selection of only one FAD seems to be a common observation. This suggests that even when AFADs are distant by more than 35 km, the competition between two adjacent devices could occur. This hypothesis must be investigated, in particular through new experiments, using echosounder buoys [25, 47] attached to AFADs. Such a protocol would allow to investigate whether adjacent FADs can simultaneously host large tuna aggregations.

Echosounder buoys attached to AFADs could also contribute to investigate the information provided by fishers that some FADs always attract more (or less) tuna than others. It is difficult to consider that design or location of FADs could explain why some FADs attract more tuna than others, as all AFADs in the Maldives are built following the same design and are anchored very deep (> 1000 m). By monitoring FAD aggregations through echosounder buoys over long periods (several weeks and months), it would be possible to further understand this information provided by fishers.

Most fishers (64.8%) believe that multiple schools of tuna, segregated by species and size, form the aggregation around AFADs. These fishers observed that during fishing operations around AFADs, when several vessels fish simultaneously, some vessels caught only small size fish while other vessel caught bigger fish. The same pattern was reported by Moreno et al, where skippers of purse seines also considered that several tuna schools form an aggregation around a DFAD, segregated by species and sizes [10]. Macusi et al, also reported a similar pattern for tuna at AFADs in the Philippines [28]. Some fishers (35.2%), however, believe that there is only one large school where different species and sizes mix. It is possible that both

situations occur at AFADs, depending on the local biomass and environmental/ oceano-graphic conditions.

All fishers observed that fish move closer to the surface during sunrise and late afternoon, increasing their catchability at the AFADs during these periods. This diel behaviour, with tuna swimming deeper during the day and shallower during the night, is well known for pelagic species, and has been observed for tuna in electronic tagging studies [28, 48, 49]. This matches the usual upward movement of the Deep Scattering Layer (DSL) at night (see Dagorn et al. [50]). A few fishers (11.1%) also thought there was some variation in the horizontal positioning of tuna at AFADs during the day. Schaefer and Fuller during their ultrasonic telemetry experiments at DFADs in 2013 observed that skipjack schools break into small sub-schools and tend to move away from the FAD but return to it after few hours [48]. However, it is possible that the fishers in the Maldives, that fish at the AFADs only during the daytime, do not experience such horizontal movements.

The time tuna aggregations spend at FADs is considered as a key parameter to characterize their associative behaviour and one of the elements to derive novel indices of abundance. Capello et al, developed a new method for estimating the local abundance of tropical tuna that uses the characteristics of the FAD associative behaviour of tuna, the time fish spend at FADs being one of the parameters used in the model [51]. It is worthy to note the differences between the residency times of individual fish and those of fish aggregations. The residency of individuals and aggregations at FADs are obviously linked, but could be different, as aggregations could result from the turn-over of individuals: some fish can join the aggregation while others leave and the aggregation can be maintained. Two methods exist to measure the time fish aggregations stay at FADs: LEK and echosounder buoys (see Baidai et al. [52]). Maldivian fishers provided a very wide range of times tuna aggregations stay at aFADs, suggesting that fish can spend very little time at FADs (30% of fishers considered that tuna stay less than 3 days at aFADs), or can stay more than 10 days (9% of answers), with all intermediate situations. In the study by Macusi et al, in the Philippines, fishers also provided a wide range of times (from one week to more than one month) for tunas to stay around FADs [28]. Nearly half of the fishers interviewed believed that tuna stay at AFADs for 2 weeks, which is considerably longer than what is reported by Maldivian fishers. Recently, estimated time series of presence/absence of tunas at DFADs using echosounder buoys they found that in the Indian Ocean, DFADs were continuously occupied by tuna aggregations for 6 days in average [52]. These values are of the same order than those provided by Maldivian fishers. Several acoustic tagging studies were designed to document the time individual tuna spend associated with AFADs and DFADs, including a study in the Maldives, which could directly be compared to our study.

In the Maldives, individual tuna were observed to stay in average less than a day or 3 to 4 days associated to FADs, depending on the period and the species, but the maximum observed residency was 12.8 days [29]. Information provided by fishers are coherent with these field results. Similar studies conducted in other AFAD arrays showed average FAD residency times from 2 to 10 days, depending on the species and area [25, 26]. Pérez et al, compared FAD residency times of tuna between aFAD arrays differing by their inter-FAD distances (or FAD densities) [36]. They found that the durations of associations of tuna at aFADs in the Maldives were shorter than those measured at aFADs in Mauritius and Hawaii. They concluded that when inter-FAD distances decrease, fish spend more time associated with FADs, suggesting that this could be a result of social behaviour and/or prey availability. As Maldivian fishers regularly visit aFADs, we believe that they could note and report more precise information on the time when aFADs are occupied by tuna (and vice versa, when they are empty). By involving fishers in the collection of such new information, we could obtain time series that could be

compared with data from echosounder buoys attached to aFADs in order to evaluate the effectiveness of such a protocol. This would have both advantages: better involvement of local fishers into science, and then management, and cheaper collection of data on the behaviour of tuna at AFADs (as compared to electronic tagging or echosounder buoys).

Most of the fishers (74.1%) suggested that tuna could be attracted to AFADs as from 2 miles (about 3.7 km) while some fishers suggested that the attraction distance varies depending on the area and the season. Skippers of tuna purse seiners considered that tuna could be attracted towards DFADs from 0–5 nautical miles (0–9 km) [10, 27]. Data collected from active acoustic tracking of tuna at AFADs in the Indian and Pacific Oceans [50, 53–55] suggest that the orientation distances ranged from 4 to 19 km, with a mode at about 10 km. This was similar to the estimates of the skippers of the purse seiners but the attraction distance suggested by Maldivian fishers are lower. The distance proposed by the Maldivian fishers could be influenced by the visible sighting distance of the AFAD buoy (about two miles radius)–beyond which the buoy is often not visible to eyes. Active tracking of tuna at AFADs in the Maldives would document the attraction distance more precisely, but it is noteworthy that all studies, including ours, tend to suggest that tuna could be attracted from a few kilometers to AFADs.

Most fishers (98.3%) believed that more than a single factor contributes towards the formation and dissolution of tuna aggregations at AFADs. A large majority agreed that the currents had the highest impact on the aggregations at AFADs. Skippers of large tropical tuna purse seiners also reported that strong currents, changes in temperature and rough sea conditions can have a strong impact on the tuna aggregations at FADs [10]. Macusi et al, also identified changes in sea currents as one of the main reasons for tuna departure from AFADs in the Philippines [28]. Several other studies have also suggested that the residency of tuna at FADs, and their departure, could be influenced by local oceanographic conditions [54, 56, 57].

Some fishers (46.3%) believed that the presence of prey help form aggregations while (40.7%) believed absence of prey cause the aggregations to leave the FAD. These fishers believed that tuna feed on various prey items in the vicinity of the AFADs and the abundance of tuna is related to local prey abundance. Several studies have suggested that prey availability around the FADs affects the duration of the aggregations [23, 58, 59]. In the pole-and-line tuna fishery of the Maldives, fishers almost daily chum large quantities of livebait at the AFADs and some fishers suggested that this could encourage the tuna to remain at the AFADs longer but CRTs in the Maldives are no longer than in other countries. Most fishers (75.9%) did not believe that large school size leads to tuna departures from the AFADs. This is also similar to the observations made by the purse seine fishers [27].

Some fishers (48.1%) also suggested that large predators such as sharks, dolphins and toothed whales can affect the behaviour of tuna. Fishers (18.5%) consider that sharks associated with AFADs could somehow help maintain tuna aggregations at AFADs. This striking information has never been documented in any other scientific publications. Forget et al, and Filmalter et al, using acoustic tracking, showed that silky sharks made excursions away from a DFAD, being closely associated with a school of tuna [49, 60]. While these studies indicate that silky sharks and tunas tend to exhibit similar associative patterns, they do not allow to investigate whether sharks play a role in the associative behaviour of tuna to floating objects. Fishers (29.6%) also observed that continuous chasing of tuna by predators negatively affected catchability of tuna. The same effect of marine mammals on the departure of tuna from FADs was reported both by fishers fishing on AFADs in the Philippines [28] and by skippers of purse seiners fishing on DFADs [27]. Generally, these reports tend to suggest that the presence of predatory mammals has a negative impact on the residency of tuna at FADs.

Maldivian fishers believe that attractants such as additional ropes, floats and netting on AFADs also influence the tuna aggregations. The AFADs deployed in the Maldives have a set of small floats weaved together as an attractant. Sometimes fishers attach pieces of thick ropes or occasionally, pieces of nets that they have recovered from DFADs to serve as attractants. Tuna fishers regularly check on the attractants attached to the AFADs and if the attractants are missing from an AFAD, they inform MoFMRA to attach new attractants. They believe these attractants provide some form of shelter to the small fish that help to form tuna aggregations. However, studies have investigated the diet of tuna associated with floating objects and concluded that tuna do not feed extensively on associated fauna [61]. Small fish associated to AFADs could therefore play a role in the association of tuna through other mechanisms, such as production of signals (e.g. sounds) which could help tuna locate FADs.

## Conclusion

With an increase in demand for tuna, both locally and internationally, it is important to ensure the sustainability of the fishery. The sustainability of the fishery depends on the status of the fish resources, the health of the ecosystem, the livelihoods of fishers. Assessing fish resources and the health of the ecosystem requires a good understanding of the marine ecosystem, including the behaviour of tuna. For instance, Capello et al, developed a method to derive indices of abundance from characteristics of the FAD associative behaviour of tuna [51]. In a more general way, the behaviour of fish, including seasonal variations in the abundance, along with the biology of fish and fishery statistics (e.g. catch and effort), are necessary for assessing the health of the resources. Knowledge on the behaviour of fish is commonly collected through scientific methods (e.g. tagging, sonars, etc.) but fishers, through other methods, also developed knowledge on fish behaviour. Scientists should combine information from scientific methods as well as from LEK. Another major output of this study corresponds to the involvement of local fishers in science and subsequently in the management of the fishery. We recommend to regularly conduct LEK studies (e.g. every year), instead of punctual and ephemeral ones, for two main reasons. First, it provides a regular flow of information allowing for time series, always useful to monitor and understand the evolution of a system. Of course, questions should be adapted to knowledge that can change every year. Second, it keeps fishers involved in science, realizing that their knowledge is valuable and used by scientists. This appears to be important to close the gap between fishers and scientists, which can also contribute to close the gap between fishers and managers.

## Supporting information

**S1 Appendix.**
(DOCX)

**S1 Data.**
(CSV)

## Acknowledgments

At foremost authors thank the Maldivian tuna fishers who contributed to this study by sharing valuable information on tuna fishing at AFADs. We also thank the two reviewers and the editor for their valuable comments that helped improve the manuscript. Maldives Marine Research Institute of the Ministry of Fisheries, Marine Resources and Agriculture (MoFMRA) was also instrumental in facilitating the field work.

## Author Contributions

**Conceptualization:** Manuela Capello, Laurent Dagorn.

**Data curation:** Ahmed Riyaz Jauharee, Monique Simier.

**Formal analysis:** Monique Simier.

**Funding acquisition:** Mohamed Shiham Adam.

**Investigation:** Ahmed Riyaz Jauharee.

**Methodology:** Manuela Capello, Laurent Dagorn.

**Project administration:** Laurent Dagorn.

**Supervision:** Manuela Capello, Laurent Dagorn.

**Writing – original draft:** Ahmed Riyaz Jauharee.

**Writing – review & editing:** Manuela Capello, Fabien Forget, Mohamed Shiham Adam, Laurent Dagorn.

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
