## [Decision Letter · Decision Letter 0]

30 Jul 2020

PONE-D-20-15145

Tuna behaviour around anchored FADs inferred from Local Ecological Knowledge (LEK) of pole-and-line tuna fishers in the Maldives

PLOS ONE

Dear Dr. Jauharee,

Thank you for submitting your manuscript to PLOS ONE. After careful consideration, we feel that it has merit but does not fully meet PLOS ONE’s publication criteria as it currently stands. Therefore, we invite you to submit a revised version of the manuscript that addresses the points raised during the review process.

INTRODUCTION:

The manuscript needs more work in the introduction and discussion sections mainly to make the reader understand how the knowledge acquired in this study could contribute to the management of tuna caught around AFAD arrays.  It would be good to have a specific section on this, to better understand the management and the fleet segments that compose the fishery fishing with AFADs.

The paper requires a detailed characterization of the existing fisheries in Maldives, including tuna fisheries, gears used as well as other subsistence fisheries that in one way or another is still dependent on FADs.

The introduction needs to introduce to the theories that are often used in tuna tagging, such as Soria's paper on the meeting point hypothesis? Or Jacquemet's paper on the influence of tuna prey.

The paper would also benefit from establishing a link between the research questions with their usefulness for the management of tuna in Maldivian waters. Additionally, in the discussion section I would suggest that authors include an insight into how fisher’s knowledge could be used systematically for knowledge production for tuna management.

METHODS:

The reviewers (and the editor) raised some issues with the wording / options available to fishers, as part of the questionnaire.  While you cannot revise the questions after the fact, please address these comemnts and concerns in the discussion.  If any of the questions are problematic, you may remove them from the analysis.

Because multiple fishers of different “status levels” were samples from the same boats, their observations may not be independent. I urge you to consider these paired observations, by performing tests where you compare the data from different people from the same vessel (e.g., captains versus crew).  You could easily calculate the % of the answers in agreement of pairs of people when they come from the same or from different boats. 

It is not clear if both artisanal or commercial fishers are considered in the interviews. If both were taken into account, please specify the percentage are artisanal and commercial fishers interviewed.  Also, you may want to compare these answers of the two groups.

Please explain why the PCA approach was used and provide examples where this method has been used in similar analyses previously.

The PCA analysis and reporting needs more details, and improvements in the figures and in the results reported.  For instance:  was PCA based on a correlation or a variance matrix, what was the total amount of pattern (variance) explained, how was the 2- axis solution selected, how many axes were considered in the analysis, were the data relativized before the analysis, what are the correlations of the variables to the PC axes, what are the loadings of the variavble3s to the PC axes, and finally – PCA requires normality and the lack of outliers; yet – there are no assessment of these critical assumptions.  Please note that a NMDS (nonparametric ordination) could be used instead.

The comparisons of the groups after the PCA should be based on a permanova or a multi-response permutation procedure test?   Please provide references and explain how this approach is done: basically, calculating the distances from the points to the origin using randomization tests.  Moreover, please explain how the p value was calculated and whether tests involving more than 2 groups were determined on the basis of all possible pair-wise tests or on the basis on an omnibus test.   Finally, please provide a table with the randomization results, showing the empirical test statistic, the distribution of randomized test statistics, and the p value calculation.

There was also a mention of logbooks in the methods section, but this was not shown or mentioned in the results.  When comparing the logbook data (visits) to the east and west, please perform a contingency test to show whether this is in fact statistically significant.

RESULTS AND DISCUSSION:

The discussion needs to relate the results to the theories that are often used in tuna tagging and how they confirm that or refute data from other sources.  Please make sure the ideas raised in the introduction are revisited in the discussion.

The paper uses vague terms like “most” and “some”.  Please provide specific %s to back up these points.

The tables and figures often do not add to 100%.  Please make sure you explain why this is the case.

Figure 1: please add a scale bar, and a line delineating the north and south regions

Figure 3: please add a scale bar, and a line delineating the east / west (and central) regions

Figure 4: Add values to the axes (X and Y), and make sure the points belonging to different groups can be easily compared.  I would also remove the lines connecting the points to the origin.

In addition to these comments, I refer you to the reviewer suggestions and to the two annotated copies of the ms, enclosed with this review.   I included some editorial suggestions and comments (highlighted in yellow and annotated) directly in the ms.

We look forward to receiving your revised manuscript.

Kind regards,

David Hyrenbach, Ph.D.

Academic Editor

PLOS ONE

Journal Requirements:

2.We note that you have indicated that data from this study are available upon request. PLOS only allows data to be available upon request if there are legal or ethical restrictions on sharing data publicly. For more information on unacceptable data access restrictions, please see http://journals.plos.org/plosone/s/data-availability#loc-unacceptable-data-access-restrictions.

3.Thank you for stating the following in the Funding Section of your manuscript:

[This study was fully funded by International Pole & Line Foundation (IPNLF)]

 [This manuscript is part of the PhD study by the lead author, A R Jauharee. The funders had no role in study design, data collection and analysis, decision to publish, or preparation of the manuscript.)]

Reviewers' comments:

Reviewer's Responses to Questions

**Comments to the Author**

1. Is the manuscript technically sound, and do the data support the conclusions?

Reviewer #1: Partly

Reviewer #2: Yes

2. Has the statistical analysis been performed appropriately and rigorously? 

Reviewer #1: Yes

Reviewer #2: Yes

3. Have the authors made all data underlying the findings in their manuscript fully available?

Reviewer #1: Yes

Reviewer #2: Yes

4. Is the manuscript presented in an intelligible fashion and written in standard English?

Reviewer #1: Yes

Reviewer #2: Yes

5. Review Comments to the Author

Reviewer #1: This original piece of research uses the knowledge of pole and line fishers to understand tuna associative behavior around anchored FADs. The manuscript is well written and easy to understand, the data and methods used are accurate. Given the importance of this fishery not only for Maldivian economy but for the global catch of the pole and line tuna fishery, the topic and the results are of primary importance. I find specially interesting the use of local ecological knowledge that has proven to be very useful in knowledge production to manage fisheries but that still is poorly or not considered in tuna Regional Fisheries Management Organizations (RFMOs).

I think the manuscript needs more work in the introduction and discussion sections mainly to make the reader understand how the knowledge acquired in this study could contribute to the management of tuna caught around AFAD arrays. I would suggest linking the research questions with their usefulness for the management of tuna in Maldivian waters. Also, in the discussion section I would suggest that authors include an insight into how fisher’s knowledge could be used systematically for knowledge production for tuna management.

Some other suggestions/comments:

Introduction

Apart from the comment above I would suggest providing within the introduction information on:

1.- The management of AFADs by fishers. Who are the owners? How are arrays of AFADs maintained?

2.- More details on the fishery and fishing strategy with AFADs. Number of AFADs visited per trip, are AFADs visited by more than one vessel simultaneously? any kind of collaboration between fishers? number of fishing companies, are the captains the ship-owners? Are there big companies with many vessels? What are data reporting requirements?

It would be good to have a specific section on this, to better understand the management and the fleet segments that compose the fishery fishing with AFADs.

Line 67: what are the percentages of catches from logs, DFADs, AFADs, dolphins.. etc.? it would be good to specify it.

Line 71: thanks for providing the local names as “oivaali”. Those local terms are important to understand the importance of AFADs for fishers and their understanding of the fishery. They should not be forgotten.

Line 80. Pole and line and hand-line? What are other fishing gears used out of the 3 nautical miles radius? If tuna are associated to AFADs beyond 3 nautical miles, this means that there are other fishing gears fishing on tuna associated to FADs? just a thought.

Line 82. What is the importance of the low density of AFADs in Maldives, in relation to other FAD arrays in other oceans/ regions? What biological or economic significance has this specific feature?

Line 105. As commented above, I would suggest developing a bit more the importance of these research questions in relation to management and perhaps in relation to what it is already known, through experiments at sea, of tuna´s associative behavior with AFADs in Maldives.

Material and Methods

Line 113. Review the sentence, “From the north..” of what? Up to or down to?

Line 120. It is not clear if for the interviews, artisanal or commercial fishers are considered, or if both are taken into account. If the latter, in which percentage are artisanal and commercial fishers interviewed? I understand the knowledge may be different as commercial fishers usually operate with technology that artisanal fishers do not have and thus, this would allow commercial fishers a greater observation capacity. The knowledge of the 2 types of fishers may be different/complementary.

L135. Is there any other characteristics other than “years in the fishery” that could be helpful to select the fishing experience or the appropriate experts? My understanding about fishers is that not always the “years in the fishery” or the age of the fisher guaranties their expertise (as in research). Other characteristics as being recognized as “knowledgeable” among fishers may be more important than the “years at sea”. For future studies, it would be good to use a method to select fishers, other than the years spent at sea.

A useful reference: Davis, A., and Wagner, J. 2003. Who knows? On the importance of identifying “experts” when researching local ecological knowledge. Hum. Ecol. 31: 463-489

L144. I see that different choices were proposed to fishers to specific questions. I would suggest for future studies not directing their answers by providing options, as these options will probably close the potential for a free flow of unbiased information from fishers. The open-ended questions would also allow the identification of new knowledge.

Results

L183. Table 2. I would suggest changing question 2 to something like: “number of days that the tuna aggregation is retained or remains at an AFAD”. If I´m not wrong, there is no way to know the residence time of an individual tuna at a given AFAD, unless they are observed one by one (tagging). So that, I would say fishers´ knowledge would be on the entire aggregation.

L227. How do fishers know those attraction distances? Which tools/knowledge were used by fishers to know that a given tuna is attracted from 5 miles away to an AFAD? Did authors test the quality of a given observation? By asking “why and how do you know this?” there is nothing on this on material and methods.

L230. I find “slight” and “strong” currents too unprecise. Authors should specify or should have asked for more detailed data (range of current speeds). This information may be very interesting to study the biomass associated related to different current speeds and directions.

Discussion

L249. I would move and mention the tagging study in the introduction.

L298. Remove “and” at the end of the sentence or add maybe environmental/oceangraphic conditions?

L299. I would suggest further discussing here and in the discussion in general, the potential disturbance of the natural behavior of tuna aggregations, provoked by pole and line fishers feeding with live bait those aggregations.

L310. As said before and as explained later by authors, I would say that fishers can´t know the time spent by a single tuna but they do know about the entire aggregations, thus, I would add “The time tuna aggregation spends”…

L 361. I would suggest detailing the strategy with the live-bait in the introduction.

L395. It would be good to provide some insight into the way this information could be gathered systematically as mentioned at the beginning of this review.

Reviewer #2: I think what the paper lacks is more on characterization of the existing fisheries in Maldives, including tuna fisheries, gears used as well as other subsistence fisheries that in one way or another is still dependent on FADs. I believe that the case for Indonesia, the Philippines as well as other tropical fisheries, they are multigear and multispecies. Although it is possible to be more selective as in the case of skipjack tuna for the pole and line fishing. However I find that the introduction lacks this basic description of the fisheries as well as the description of the study site is also lacking in more details, including what made you decide to use a PCA to relate the 54 fishers, the type of schools, and the species of tuna. Further on, the discussion needs to relate more to the theories that are often used in tuna tagging and how they confirm that or not, for instance Soria's paper on the meeting point hypothesis? Or Jacquemet's on tuna prey...I think the paper should also relate to these ones.

6. PLOS authors have the option to publish the peer review history of their article (what does this mean?). If published, this will include your full peer review and any attached files.

Reviewer #1: No

Reviewer #2: No

---

## [Author Response · Author response to Decision Letter 0]

24 Apr 2021

David Hyrenbach, PhD,

Academic Editor

PLOS ONE

September 07, 2020

Dear Dr Hyrenbach,

Subject: Submission of revised manuscript

Thank you for your email dated July 30, 2020 enclosing the editor and reviewer comments. The comments were extremely helpful in improving the quality of the manuscript. We really appreciate all the efforts made to help us improve the manuscript. Please find our responses to the queries of the reviewers and editor below. 

Yours sincerely,

A Riyaz Jauhree

RESPONSES TO THE EDITOR COMMENTS

A – Responses to general comments from the email

Comment 1: 

The manuscript needs more work in the introduction and discussion sections mainly to make the reader understand how the knowledge acquired in this study could contribute to the management of tuna caught around AFAD arrays. It would be good to have a specific section on this, to better understand the management and the fleet segments that compose the fishery fishing with AFADs. 

The paper requires a detailed characterization of the existing fisheries in Maldives, including tuna fisheries, gears used as well as other subsistence fisheries that in one way or another is still dependent on FADs.

Response:

The manuscript has been modified accordingly to better describe the tuna fishery and its management in the Maldives, with a separate section on “The Maldivian tuna fishery and its management”. 

Comment 2: 

It is not clear if both artisanal or commercial fishers are considered in the interviews. If both were taken into account, please specify the percentage are artisanal and commercial fishers interviewed. Also, you may want to compare these answers of the two groups.

Response:

In this study only commercial pole and line fishers were interviewed. A sentence has been modified accordingly in the section “fishers interviews” in the Materials and Methods:

“The fishers for the interviews were selected based on their fishing experience and area of fishing. Hence captains (n=36), deputy captains (n=9) and crew members (n=9) from 36 vessels, who had a minimum of 8 years of experience on a licensed commercial pole-and-line vessel in the Maldives were selected.”

Comment 3: 

Please explain why the PCA approach was used and provide examples where this method has been used in similar analyses previously.

Response:

We have detailed the PCA approach used with additional references. 

“The responses were coded “1” for “yes” and “0” for “no” and were analysed by developing a data table with 12 columns (the months) and 540 rows (the answers of the 54 fishers for the 5 school types and the 2 species). Multidimensional data from LEK questionnaires are often analysed using multivariate approaches such as nDMS and PERMANOVA [39], Multiple Correspondence Analysis [8], and Principal Component Analysis [40]. Here, the table was subjected to a Principal Component Analysis on covariance matrix (centered PCA) in order to obtain an overview of the seasonality of tuna abundance, in relation with the species and the type of school but also with the origin and the position of the fishers. Four between-groups PCAs, a particular case of PCA with respect to instrumental variables [41] in which there is only a single factor as explanatory variable, were then performed to compute the ratio of variability explained by (i) the position of fishers (captains, deputy captains and crew members), (ii) the origin of fishers (north and south), (iii) the different types of schools and (iv) the species (skipjack and yellowfin). Monte-Carlo permutation tests [42] with 1000 random permutations were finally conducted to assess the significance level of the observed ratios in between-groups analyses.”

Comment 4: 

There was also a mention of logbooks in the methods section, but this was not shown or mentioned in the results. When comparing the logbook data (visits) to the east and west, please perform a contingency test to show whether this is in fact statistically significant.

Response:

This information is now included in the results section.

“The mid-day positions of pole-and-line fishing vessels obtained from logbooks during the northeast and southwest monsoons (Fig 3) showed a similar trend, with fishers tending to fish more on the east (45% - fishing events) than on the west (32% - fishing events) of the Maldives during the northeast monsoon (Chi-2 test p-value<0.0001) and on the west (42% - fishing events) than on the east (39% - fishing events) of the Maldives during the southwest monsoon (Chi-2 test p-value<0.009). The remaining vessels reported the central region as their mid-day position.”

... and in the Material and methods section:

“Chi-2 tests were used to compare the number of visits reported in the logbook between East and West of the Maldives.”

Comment 5: 

The introduction needs to introduce to the theories that are often used in tuna tagging, such as Soria's paper on the meeting point hypothesis? Or Jacquemet's paper on the influence of tuna prey.

Response:

A section on the hypotheses to explain why tuna associate with FADs was added in the Introduction: 

“Several pelagic species, including tropical tuna, are naturally attracted to floating objects such as drifting logs or marine debris [11]. Although several hypotheses have been postulated to explain the associative behaviour of tuna [12], including the meeting point hypothesis [13,14] and the indicator-log hypothesis [15], we still do not know why tuna associate with these objects. Naturally, this has not prevented fishers from taking advantage of this particular behaviour to help them find and catch tuna. The Roman author Oppian (200 AD) first reported catches of pelagic fish around floating objects in the Mediterranean sea, while aggregating devices were used in Japan in the 17th century [16]. The use of floating objects by fishers developed considerably in the 20th and 21st centuries with the expansion of fisheries targeting tropical tuna and the use of Fish Aggregating Devices (FADs).”

Comment 6: 

The paper would also benefit from establishing a link between the research questions with their usefulness for the management of tuna in Maldivian waters. Additionally, in the discussion section I would suggest that authors include an insight into how fisher’s knowledge could be used systematically for knowledge production for tuna management.

Response:

We added a Conclusion, where we recommend the regular use of LEK to collect information on the behaviour of fish, and keep fishers involved in science. 

METHODS:

Comment 7: 

The reviewers (and the editor) raised some issues with the wording / options available to fishers, as part of the questionnaire. While you cannot revise the questions after the fact, please address these comments and concerns in the discussion. If any of the questions are problematic, you may remove them from the analysis.

Because multiple fishers of different “status levels” were sampled from the same boats, their observations may not be independent. I urge you to consider these paired observations, by performing tests where you compare the data from different people from the same vessel (e.g., captains versus crew). You could easily calculate the % of the answers in agreement of pairs of people when they come from the same or from different boats. 

Response:

We did not further compare the responses between the captains and the crew members from the same boat as only 9 interviews were conducted allowing to test the “vessel” effect, which is very low. 

Comment 8: 

The PCA analysis and reporting needs more details, and improvements in the figures and in the results reported. For instance: was PCA based on a correlation or a variance matrix, what was the total amount of pattern (variance) explained, how was the 2- axis solution selected, how many axes were considered in the analysis, were the data relativized before the analysis, what are the correlations of the variables to the PC axes, what are the loadings of the variables to the PC axes, and finally – PCA requires normality and the lack of outliers; yet – there are no assessment of these critical assumptions. Please note that a NMDS (nonparametric ordination) could be used instead.

The comparisons of the groups after the PCA should be based on a permanova or a multi-response permutation procedure test? Please provide references and explain how this approach is done: basically, calculating the distances from the points to the origin using randomization tests. Moreover, please explain how the p value was calculated and whether tests involving more than 2 groups were determined on the basis of all possible pair-wise tests or on the basis on an omnibus test. Finally, please provide a table with the randomization results, showing the empirical test statistic, the distribution of randomized test statistics, and the p value calculation.

Response: (see also response to Comment #12 below)

The PCA was applied on the covariance matrix (Centred PCA) because all variables (months) are in the same unit and the data didn’t need to be relativized for this reason. The use of covariance matrix allowed us to identify the months with globally low tuna presence according to the fishers (e.g. June and August) and the months with high tuna presence (e.g. November, December…).

The first two axes were selected because they summarize 34% of the total variance of the table: “The first two axes of the PCA (Fig 4) summarized respectively 20% and 14% of the information…” (Result section). The 2 first axes are by definition the best summary of the information.

When PCA is used (as done here) for exploratory purposes, normality is not a strict requirement. The algorithm used here, from ade4 library is based on a purely geometrical technique (Dray & Dufour, 2007) which does not require normality. We tried before a Correspondence Analysis, but the results did not show clear seasonality. We also tried a NDMS but it did not converge. 

The between-groups PCA is a method to compute the ratio of variance of the data table explained by a categorical variable, in the general framework of Principal Component Analysis with respect to Instrumental Variables, as explained by Lebreton et al, 1991 (in the reference list). The significance level of the ratio of explained variance was test for each categorical variable using Monte-Carlo permutation tests (Romesburg, 1985). 

Relevant modifications were made in the Material & methods and result sections to include these information. 

“The responses were coded “1” for “yes” and “0” for “no” and were analysed by developing a data table with 12 columns (the months) and 540 rows (the answers of the 54 fishers for the 5 school types and the 2 species). Multidimensional data from LEK questionnaires are often analyzed using multivariate approaches such as nDMS and PERMANOVA [39] Multiple Correspondence Analysis ([8], and Principal Component Analysis [40]. Here, the table was subjected to a Principal Component Analysis on covariance matrix (centered PCA) in order to obtain an overview of the seasonality of tuna abundance, in relation with the species and the type of school but also with the origin and the position of the fishers. Four between-groups PCAs, a particular case of PCA with respect to instrumental variables [41] in which there is only a single factor as explanatory variable, were then performed to compute the ratio of variability explained by (i) the position of fishers (captains, deputy captains and crew members), (ii) the origin of fishers (north and south), (iii) the different types of schools and (iv) the species (skipjack and yellowfin). Monte-Carlo permutation tests [42] with 1000 random permutations were finally conducted to assess the significance level of the observed ratios in between-groups analyses.”

RESULTS AND DISCUSSION:

Comment 9: 

The discussion needs to relate the results to the theories that are often used in tuna tagging and how they confirm that or refute data from other sources. Please make sure the ideas raised in the introduction are revisited in the discussion.

The paper uses vague terms like “most” and “some”. Please provide specific %s to back up these points.

The tables and figures often do not add to 100%. Please make sure you explain why this is the case.

Response:

Appropriate modifications were made to address this. Specific percentage values are now included in the statements. 

Comment 10: 

Figure 1: please add a scale bar, and a line delineating the north and south regions

Response:

Changes were made as per the suggestion.

Comment 11: 

Figure 3: please add a scale bar, and a line delineating the east / west (and central) regions.

Response:

Changes were made as per the suggestion.

Comment 11: 

Figure 4: Add values to the axes (X and Y), and make sure the points belonging to different groups can be easily compared. I would also remove the lines connecting the points to the origin.

Response:

Changes were made as per the suggestion.

 

B – Responses to Editor comments made on the word document.

Comment 1: 

Line #41: Can you please provide more context: they move to the surface to forage?

Response: 

As the reviewer requested, we have improved the discussion on this result in the Discussion:

“All fishers observed that fish move closer to the surface during sunrise and late afternoon, increasing their catchability at the AFADs during these periods. This diel behaviour, with tuna swimming deeper during the day and shallower during the night, is well known for pelagic species, and has been observed for tuna in electronic tagging studies (Schaefer and Fuller 2013 ; Forget et al. 2015 ; Macusi et al. 2017). This matches the usual upward movement of the Deep Scattering Layer (DSL) at night (see Dagorn et al. 2000)”. 

Comment 2: 

Line #53: Add scientific name

Response: 

The scientific names have been added:

“About 75% of the catch, mainly skipjack (Katsuwonus pelamis) and small yellowfin (Thunnus albacares) (FL<65cm) tuna, is caught using pole-and-line [3], the other main gear being handline”.

Comment 3: 

Line #54: Please clarify: is this standard length or fork-tail length?

Response: 

It is fork length. This has been clarified in the sentence:

“About 75% of the catch, mainly skipjack (Katsuwonus pelamis) and small yellowfin (Thunnus albacares) (Fork Length < 65 cm) tuna, is caught using pole-and-line [3], the other main gear being handline”.

Comment 4: 

Line #62: How about: “travel north pursuing better fishing conditions”.

Response: 

We have modified the sentence accordingly:

“Most of these commercial pole-and-line vessels operate in the south of the Maldives can travel north to pursue better fishing conditions.”

Comment 5: 

Line #109:

IS ITEM #2 NOT ALREADY PART OF ITEM #1 AND #3?

Response: 

The last sentence of the Introduction has been modified accordingly:

The objective of this study is to use LEK to improve knowledge of tuna behaviour at AFADs in the Maldives, at the scale of the FAD array (including seasonal variations) and at the scale of individual AFADs.

Comment 6: 

Line #114: CAN YOU PLEASE PROVIDE A BRIEF DESCRIPTION OF THE SURFACE CURRENTS? ALSO, PLEASE ADD THESE TO FIGURE 1. YOU COULD EVEN PROVIDE TWO MAPS: FOR THE NE AND THE SW MONSOON CONDITIONS.

Response: 

We provided a reference which provides a great deal of details on the surface currents of the Indian ocean and its seasonal variability.

Schott FA, McCreary JP. The monsoon circulation of the Indian Ocean. Prog Oceanogr. 2001;51: 1–123. doi:10.1016/S0079-6611(01)00083-0

Figure 1 was improved to make the current symbols clearer. 

Comment 7: 

Line #118: NOTE: IF THE BATHYMETRY IS IMPORTANT, CAN YOU PLEASE PROVIDE SOME DEPTH CONTOUTRS IN FIGURE 1?

Response: 

Considering that all AFADs are anchored deeper than 1000 m, bathymetry is not important for the purpose of the current study, except the contours of the atolls, represented in figure 1. 

Comment 8: 

Line #120: PLEASE NAME THE SURFACE CURRENTS AND BRIEFLY EXPLAIN IF THERE IS EQUATORIAL UPWELLING ASSOCIATED WITH THEM. THANKS.

Response: 

Additional information on currents were added to figure 1 with two different symbols representing the two monsoon currents.

Comment 9: 

Line #122 (#133 in revised document): CAN YOU PLEASE PROVIDE SOME STATISTICS: LIKE THE MEAN +/- SD OF THE ANNUAL CATCHES. ALSO, PLEASE PROVICE MORE DETAILS: WHAT YEARS WERE THESE STATISTICS REFERING TO. THANKS.

Response: 

Catch statistics have been included in the text as suggested in the Materials and Methods:

(Average pole-and-line catch for the last 5 years: North (7° to 2°N) 16,000t (SD±1000t) and South (2°N to 1°S) 49,500t (SD±6000t). (MoFMRA, 2019)).

Comment 10: 

Line #142 (#153 in revised document): PLEASE EXPLAIN WHY? YOU MENTIONED SOME OF THE PHYSIOGRAPHIC DIFFERENCES IN THE INTRODUCTION

Response: 

The physiographic differences of the north and south regions of the Maldives are described in the study site. Additional content has been added for explanation for why 2 degrees north was selected as the line dividing north and south:

“For this study, the Maldives archipelago was divided into north and south area at 2.0oN (Fig 1) based on the physiographic differences of the study site [24,25] and fishing practices of the local tuna fishers.”

Comment 11: 

Line #170: DID YOU ASK THE FISHERS ABOUT FISHING IN THE NORTH / SOUTH REGIONS?

Response: 

Yes – we did ask the fishers about fishing in the north and south region but only few fishers that took part in the interview from the south have travelled north seeking better fishing grounds. None of the fishers from the north that took part in the interview travelled so far south (away from their home port) to fish.

Comment 12: 

Line #180 (#200 in revised document): NEED TO PROVIDE MORE INFORMATION ABOUT THIS ANALYSIS:

- WAS THE ABUNDANCE A CATEGORICAL OR A CONTINUOUS VARIABLE?

- WERE THE DATA RELATIVISED IN ANY WAY?

- DID YOU USE A CORRELATION OR COVARIANCE MATRIX?

FURTHERMORE, PCA ASSUMES NORMALITY AND THE LACK OF OUTLIERS. THEREFORE, DID YOU PERFORM ANY ANALYSES BEFORE THE PCA, TO ENSURE THESE ASSUMPTIONS WERE MET?

Response: 

The answers of the fishers to this question were multi-dimensional: 12 month, 2 species and 5 school types. To keep all the information, we decided to organize a synthetic table with the 12 months as variables (columns of the table), while the answers of the 54 fishers for the 2 species and 5 types of school were individuals (540 rows). This table contained 1 or 0 values, according to the positive (1) or negative (0) answers of each fisher for each species and each type of school about the month when tuna are abundant. 

The PCA was applied on the covariance matrix (Centred PCA) because all variables (months) are in the same unit and the data didn’t need to be relativized for this reason. The use of covariance matrix allowed us to identify the months with globally low tuna presence according to the fishers (e.g. June and August) and the months with high tuna presence (e.g. November, December…). 

When PCA is used (as done here) for exploratory purposes, normality is not a strict requirement. The algorithm used here, from ade4 library is based on a purely geometrical technique (Dray & Dufour, 2007) which does not require normality. We tried before a Correspondence Analysis, but the results did not show clear seasonality. We also tried a NDMS but it did not converge.

“Fishers responses on the months where tuna abundance is higher (question 1 of the questionnaire – see appendix) concerned five school types (AFAD, Log schools, DFAD, Free school, Seamount) and the two main target tuna species skipjack (Katsuwonus pelamis) and yellowfin tuna (Thunnus albacares). The responses were coded “1” for “yes” and “0” for “no” and were analysed by developing a data table with 12 columns (the months) and 540 rows (the answers of the 54 fishers for the 5 school types and the 2 species). Multidimensional data from LEK questionnaires are often analyzed using multivariate approaches such as nDMS and PERMANOVA [39], Multiple Correspondence Analysis [8], and Principal Component Analysis [40]. Here, the table was subjected to a Principal Component Analysis on covariance matrix (centered PCA) in order to obtain an overview of the seasonality of tuna abundance, in relation with the species and the type of school but also with the origin and the position of the fishers. Four between-groups PCAs, a particular case of PCA with respect to instrumental variables [41] in which there is only a single factor as explanatory variable, were then performed to compute the ratio of variability explained by (i) the position of fishers (captains, deputy captains and crew members), (ii) the origin of fishers (north and south), (iii) the different types of schools and (iv) the species (skipjack and yellowfin). Monte-Carlo permutation tests [42] with 1000 random permutations were finally conducted to assess the significance level of the observed ratios in between-groups analyses. Chi-2 tests were used to compare the number of visits reported in the logbook between East and West of the Maldives. All statistical analyses were performed using R software [43] with the ade4 package [44].”

Comment 13: 

Line #180 to 186 (#200 to 208 in revised document): IS THIS SIMILAR TOA PERMANOVA OR A MRPP TEST? WERE THE DISTANCES BETWEEN THE SAMPLES IS COMPARED AS A FUNCTIOMN OF THE GROUPING VARIABLE?

Response: 

Between-groups PCAs are a particular case of “PCA with Instrumental Variables” (see Lebreton et al., 1991), a method closely related to Redundancy Analysis (RDA) and Canonical Correspondence Analysis (CCA), which are currently used in ecology to associate faunistic and environmental or instrumental data. Between-groups PCAs are based on projected variance on a single factor. The significance level of the ratio of projected variance on the considered factor is assessed using a permutation test, i.e. a high number of random associations between rows and a level of the factor to calculate random ratios. The observed value of the ratio is then compared to the distribution of simulated ratios. We improved the description of this method as follows: 

“Four between-groups PCAs, a particular case of PCA with respect to instrumental variables [41] in which there is only a single factor as explanatory variable, were then performed to compute the ratio of variability explained by (i) the position of fishers (captains, deputy captains and crew members), (ii) the origin of fishers (north and south), (iii) the different types of schools and (iv) the species (skipjack and yellowfin). Monte-Carlo permutation tests [42] with 1000 random permutations were finally conducted to assess the significance level of the observed ratios in between-groups analyses. Chi-2 tests were used to compare the number of visits reported in the logbook between East and West of the Maldives.”

Comment 14: 

Line #195 to 200 (#216 to 223 in revised document): NOTE: THESE %S DO NOT ADD TO 100%? ARTE THE MISSING RECORDS DUE TO LACK OF REPORTING? POSITIONAL ERRORS? OR WERE THERE OTHER POSSIBLE AREAS (NORTH AND SOUTH)?

Response: 

The % do not add to 100% because the central zone (73°E – 73.5°E) is not considered here (23% for the NE monsoon and 19% for SW). New information is now added to the Materials and Methods section under logbook about the three zones – west, central and east. 

“All reported position between 70°E to 73°E were recorded as West and positions between 73.5°E to 76.5°E were recorded as East of the Maldives. Mid-day positions reported in the Central region 73°E to 73.5°E were not considered.”

Comment 15: 

Line #195 to 200: PLEASE REPORT WHETHER THESE COUNTS ARE IN FACT SIGNIFICANTLY DIFFERENT USING A CONTINGENCY TEST

Response: Chi-2 tests were performed to compare the number of visits in the East vs. West areas of the fishing zone during the NE and during the SW monsoon.

“The mid-day positions of pole-and-line fishing vessels obtained from logbooks during the northeast and southwest monsoons (Fig 3) showed a similar trend, with fishers tending to fish more on the east (45% - fishing events) than on the west (32% - fishing events) of the Maldives during the northeast monsoon (Chi-2 test p-value<0.0001) and on the west (42% - fishing events) than on the east (39% - fishing events) of the Maldives during the southwest monsoon (Chi-2 test p-value=0.009).”

Comment 16: 

Line #203 to 205 (#229 to 240 in revised document): CAN YOU PLEASE REPORT THE CORRELATIONS OF THE VARIABLES WITH THE AXES, OR CAN YOU REPORT THE LOADINGS OF THE VARIABLES IN THE VARIOUS AXES? THIS WOULD HELP INTERPRET WHAT THE AXES MEAN.

Response:

In PCA there is no need to report the correlations of the variables because they are directly related to the loadings, which are represented on Fig 4A. Axis 1 and 2 can be interpreted using this graph (4A) and this has been done in the text.

Comment 17: 

Line #226 (#253 in revised document): CAN YOU PLEASE PROVIDE A TABLE / FIGURE SHOWING THE RESULT OF THIS QUESTION.

BECAUSE YOU MERELY ASKED IF THERE WAS A DIEL CYCLE, HOW DID YOU ASCERTAIN THE CYCLES? 

DID YOU ASK FISHERS TO LABEL ACTIVITY LEVEL THROUGHOUT THE DAY? PLEASE EXPLAIN.

Response:

The response leading to this information was obtained during the discussion of question 10 – Does time of the day influence the behaviour of tuna at the AFADs? All fishers agreed that catchability is high during the early hours of the day and in the late afternoon. This information is provided in table 2. 

Comment 18: 

Line #277 (#305 in revised document): CAN YOU PLEASE PROVIDE SOME SUMAMRY STATISTICS? HOW DO THE CATCHES COMPARE BY AREA / PERIOD?

Response:

The paragraph has been rephrased and average catch statistics have been added.

“In general, there are high catches observed during the northeast monsoon (lasts for 4 months) than in southwest monsoon (lasts for 6 months). Over the last five years (2014 to 2018) MoFMRA catch statistics showed that the average catches during northeast monsoon (during a 4 months period) was 50,000±2000 t while in the southwest monsoon (during a 6 months period) it was at 54,000±3000 t.

Comment 19: 

Line # 356: HOW MANY? PLEASE PROVIDE %S?

Response:

The percentage has been added to the paragraph.

“Most fishers (98.3%) believed that more than one single …….”

Comment 20: 

Line # 365: HOW MANY? PLEASE PROVIDE %S?

Response:

The percentage has been added to the paragraph.

“Some fishers (46.3%) believed that the presence of prey affect aggregations while others (40.7%) believed absence of prey affects aggregations.”

Comment 21: 

Line #569: Figure 1 - PLEASE PROVIDE A SCALE BAR – TO SHOW THE SIZE OF THE STUDY AREA

Response:

The figure has been improved with additional information on study area.

Fig 1. The study area, AFAD network outside Maldives atolls and the direction of monsoon related currents. (Dotted arrows indicate southwest monsoon currents and the continuous arrows indication northeast monsoon currents. NMC – northeast monsoon current, SECC – south equatorial counter current, SMC – southwest monsoon current, EJ – equatorial jet).

Comment 22: 

Line #577: CAN YOU PLEASE ADD TWO VERTICAL LINES SEPARATING THE EAST AND THE WEST REGION FROM THE CENTRAL REGION. ALSO, PLEASE SHOW THE % OF VISITS FROM THE CENTRAL REGION, TO MAKE SURE %S ADD TO 100.

Response:

The vertical lines separating the regions and percentages are added to the new figure 3.

Fig 3. Mid-day position of pole-line-fishing vessels from logbook data for the SW and the NE monsoon periods for the west (70°E to 73°E), central (73°E to 73.5°E) and east (73.5°E to 76.5°E) regions.

Comment 23: 

Line #582: Figure 4 - PLEASE PROVIDE VALUES ON THE X AND Y AXES. I ASSUME THE ORIGIN IS THE SAME, BUT THE DIFFERENT PLOTS MAY NOT BE COMPARABLE – GIVEN THE DIFFERENT “D” SCALES. 

Response:

Values on the axes could be given either on the graph, or in the legend. For Fig 4A one graduation, horizontally or vertically corresponds to 0.1 (d=0.1 in the top-right corner). For Fig 4 B to E, one graduation corresponds to 0.5. 

The last sentence of the legend is modified as follows: 

“The value of d in the top-right corner gives the scale of the grid (0.1 for Fig. A and 0.5 for Fig. B to E)”

Comment 24: 

Line #582: Figure 4 - YOU MAY ALSO WANT TO EITHER REMOVE THE LINES CONNECTING THE POINTS TO THE ORIGIN, TO FACILITATE THE VISUAL COMPARISON. 

Response:

Symbols provided in the fig 4 graphs.

B – Origin: North (black square), South (circle)

C – Position: Captain (black square), Crew (triangle point down), Deputy Captain (circle plus)

D – School Type: AFADs (diamond), DFADs (circle plus), Log Schools (black triangle point up), Free schools (square), Seamounts (cross)

E – Species: SKJ (black square), YFT (circle)

Comment 24: 

Line #582: Figure 4 - ALTERNATIVELY, YOU MAY WANT TO SHOW MULTIPLE PANELS, WHERE EACH GROUP IS SHOWN SEPARATELY – FORE THE SAME COMPARISON. FOR INSTANCE: INCLUDE TWO PANELS FOR THE TWO SPECIES.

Response:

We thank the reviewer for the suggestion, however there are already 5 graphs in Fig. 4, this may result in a very cumbersome multiplot Figure. Therefore, we decided to use different symbols.

Comment 25: 

Line #582: Figure 4 - CAN YOU PLEASE PROVIDE DIFFERENT SYMBOLS, SO THE GROUPS CAN BE IDENTIFIED? FOR INSTANCE, THE TWO SPECIES CANNOT BE DISCERNED. 

Response:

Key:

B – Origin: North (black square), South (circle), C – Position: Captain (black square), Crew (triangle point down), Deputy Captain (circle plus), D – School Type: AFADs (diamond), DFADs (circle plus), Log Schools (black triangle point up), Free schools (square), Seamounts (cross), E – Species: SKJ (black square), YFT (circle). The value of d in the top-right corner gives the scale of the grid (0.1 for Fig. A and 0.5 for Fig. B to E).

Fig 4. Results of the PCA of the answers of the fishers (n=54) for the 5 types of schools and the two species (skipjack and yellowfin tuna) per month. On the first two axes, projection of the 12 months (A) and of the 540 rows grouped by origin of the fisher (B), by position of the fisher (C), by type of school (D) and by species (E). The value of d gives the scale of the grid.

RESPONSES TO REVIEWER #1 COMMENTS

Comment 1: 

I think the manuscript needs more work in the introduction and discussion sections mainly to make the reader understand how the knowledge acquired in this study could contribute to the management of tuna caught around AFAD arrays. I would suggest linking the research questions with their usefulness for the management of tuna in Maldivian waters. Also, in the discussion section I would suggest that authors include an insight into how fisher’s knowledge could be used systematically for knowledge production for tuna management.

Response: 

See response to general comment 1.

Comment 2: 

The management of AFADs by fishers. Who are the owners? How are arrays of AFADs maintained?

Response: 

The AFADs are fully owned and managed by the Maldives Government. All the costs or maintenance are borne by the Government. Changes have been done in the special section on: The Maldivian tuna fishery and its management

Comment 3: 

More details on the fishery and fishing strategy with AFADs. Number of AFADs visited per trip, are AFADs visited by more than one vessel simultaneously? any kind of collaboration between fishers? number of fishing companies, are the captains the ship-owners? Are there big companies with many vessels? What are data reporting requirements?

It would be good to have a specific section on this, to better understand the management and the fleet segments that compose the fishery fishing with AFADs.

Response:

Changes have been done in the special section on: The Maldivian tuna fishery and its management 

Comment 4: 

Line 67: what are the percentages of catches from logs, DFADs, AFADs, dolphins.. etc.? it would be good to specify it.

Response:

Unfortunately, this level of data does not exist in the logbook. We have included in the new section on Changes have been done in the special section on: The Maldivian tuna fishery and its management the percentage of catches from the field work finding.

“In the Maldives skipjack and small yellowfin tuna are caught by targeting (i) free swimming schools (45.4%), (ii) logs or other drifting objects (11.3%), (which includes drifting fish aggregating devices (DFADs) deployed by purse seiners that pass through the Maldives EEZ), (iii) seamount associated schools (10.4%), and (iv) anchored fish aggregating devices (AFADs) (32.8%). 

Reference for the percentages : Miller KI, Nadheeh I, Riyaz Jauharee A, Charles Anderson R, Shiham Adam M. Bycatch in the Maldivian pole-and-line tuna fishery. PLoS One. 2017;12. doi:10.1371/journal.pone.0177391

Comment 5: 

Line 80. Pole and line and hand-line? What are other fishing gears used out of the 3 nautical miles radius? If tuna are associated to AFADs beyond 3 nautical miles, this means that there are other fishing gears fishing on tuna associated to FADs? just a thought.

Response:

Hook and line are the only gear permitted in the Maldives. These include pole-and-line, handline, troll-line and longline. Beyond the 3 nautical mile in addition to pole-and-line handline and troll-line are used. At present longline activities have been suspended. All of this has been added in the Introduction.

Comment 6: 

Line 82. What is the importance of the low density of AFADs in Maldives, in relation to other FAD arrays in other oceans/ regions? What biological or economic significance has this specific feature?

Response:

The low density of AFADs in the Maldives is presented in the new section on Changes have been done in the special section on: The Maldivian tuna fishery and its management.

In addition, the importance of collecting information on the behaviour of tuna in FAD arrays differing by their inter-FAD distances has been discussed in the Discussion (see lines 400-412)

Comment 7: 

Line 105. As commented above, I would suggest developing a bit more the importance of these research questions in relation to management and perhaps in relation to what it is already known, through experiments at sea, of tuna´s associative behavior with AFADs in Maldives.

Response:

The objective of the paper is to improve knowledge on tuna behaviour at AFADs. Results are compared with published information on tuna behaviour in the Discussion. The importance of tuna behaviour in management is also addressed in the new Introduction and in the Conclusion

Comment 8: 

Line 113. Review the sentence, “From the north..” of what? Up to or down to?

Response:

We corrected the sentence:

“From almost the northern tip of the Maldives up to about 2.5oN, the Maldives atolls are arranged in a double chain. ” 

Comment 9: 

Line 120. It is not clear if for the interviews, artisanal or commercial fishers are considered, or if both are taken into account. If the latter, in which percentage are artisanal and commercial fishers interviewed? I understand the knowledge may be different as commercial fishers usually operate with technology that artisanal fishers do not have and thus, this would allow commercial fishers a greater observation capacity. The knowledge of the 2 types of fishers may be different/complementary.

Response:

See response to general comment 2.

Comment 10: 

L135. Is there any other characteristics other than “years in the fishery” that could be helpful to select the fishing experience or the appropriate experts? My understanding about fishers is that not always the “years in the fishery” or the age of the fisher guaranties their expertise (as in research). Other characteristics as being recognized as “knowledgeable” among fishers may be more important than the “years at sea”. For future studies, it would be good to use a method to select fishers, other than the years spent at sea.

Response:

Years of experience in the fishery is widely used in studies collecting data from fishers (Moreno et al., 2007), and this was our main criteria for selecting fishers. In addition we also looked at their seniority on the vessel. However, we agree with the reviewer that other criteria could also be used in future studies to select fishers. Catch performance for instance could be one criteria, if we consider that good fishers could also be the ones who have the best knowledge of their environment.

Comment 11: 

L144. I see that different choices were proposed to fishers to specific questions. I would suggest for future studies not directing their answers by providing options, as these options will probably close the potential for a free flow of unbiased information from fishers. The open-ended questions would also allow the identification of new knowledge.

Response:

Our strategy was to be able to quantify the information collected by fishers, which required close-ended questions. But we agree with the reviewer and will certainly consider open-ended questions in future LEK studies.

Comment 12: 

L183. Table 2. I would suggest changing question 2 to something like: “number of days that the tuna aggregation is retained or remains at an AFAD”. If I´m not wrong, there is no way to know the residence time of an individual tuna at a given AFAD, unless they are observed one by one (tagging). So that, I would say fishers´ knowledge would be on the entire aggregation.

Response:

Changes were made to the table as per suggestion, as well as in the Discussion.

Comment 13: 

L227. How do fishers know those attraction distances? Which tools/knowledge were used by fishers to know that a given tuna is attracted from 5 miles away to an AFAD? Did authors test the quality of a given observation? By asking “why and how do you know this?” there is nothing on this on material and methods.

Response:

When fishers observe tuna schools swimming in the direction of a FAD, they consider that it is attracted by it. They estimate the distance from the school to the FAD using their GPS.

Comment 14: 

L230. I find “slight” and “strong” currents too unprecise. Authors should specify or should have asked for more detailed data (range of current speeds). This information may be very interesting to study the biomass associated related to different current speeds and directions.

Response:

According to fishers who took part in the interview – strong current is above 4 knots and moderate or slight current is between 1 to 4 knots. We have added this information to the table.

Comment 15: 

L249. I would move and mention the tagging study in the introduction.

Response:

We followed the reviewer’s suggestion, by citing this tagging study in the introduction, but also further in the Discussion to discuss the results on the time fish spend at AFADs.

Comment 16: 

L298. Remove “and” at the end of the sentence or add maybe environmental/oceangraphic conditions?

Response:

Have changed accordingly

Comment 17: 

L299. I would suggest further discussing here and in the discussion in general, the potential disturbance of the natural behavior of tuna aggregations, provoked by pole and line fishers feeding with live bait those aggregations.

Response:

Of course the tuna response to bait do bring them to the surface but (1) we consider that this vertical movement is small (few meters) and (2) chumming the same bait during mid-day do not always show similar responses. 

Diel vertical pattern has been observed by other authors (Forget et al., 2015 - Behaviour and vulnerability of target and non-target species at drifting fish aggregating devices ( FADs ) in the tropical tuna purse seine fishery determined by acoustic telemetry) even when fishers do not chum.

The possible effect of chumming on the time fish spend at FADs is unknown. However, we believe that chumming does not significantly increase the time fish spend at FADs as tuna spend less times at FADs in the Maldives (where fishers chum) than in Mauritius or Hawaii (where fishers do not chum) (see Pérez et al. 2020). 

Comment 18: 

L310. As said before and as explained later by authors, I would say that fishers can´t know the time spent by a single tuna but they do know about the entire aggregations, thus, I would add “The time tuna aggregation spends”…

Response:

The reviewer is correct. Changes have been done to correct this, in particular in the Discussion (see lines 596-661).

Comment 19: 

L 361. I would suggest detailing the strategy with the live-bait in the introduction.

Response:

The strategy has been detailed in the introduction.

Comment 20: 

L395. It would be good to provide some insight into the way this information could be gathered systematically as mentioned at the beginning of this review.

Response:

A Conclusion has been added to the paper, which addresses this important comment. 

RESPONSES TO REVIEWER #2 COMMENTS

General comment: 

I think what the paper lacks is more on characterization of the existing fisheries in Maldives, including tuna fisheries, gears used as well as other subsistence fisheries that in one way or another is still dependent on FADs. I believe that the case for Indonesia, the Philippines as well as other tropical fisheries, they are multigear and multispecies. Although it is possible to be more selective as in the case of skipjack tuna for the pole and line fishing. However I find that the introduction lacks this basic description of the fisheries as well as the description of the study site is also lacking in more details, including what made you decide to use a PCA to relate the 54 fishers, the type of schools, and the species of tuna. Further on, the discussion needs to relate more to the theories that are often used in tuna tagging and how they confirm that or not, for instance Soria's paper on the meeting point hypothesis? Or Jacquemet's on tuna prey...I think the paper should also relate to these ones.

Response:

We have expanded the introduction providing a better insight into the fisheries in the Maldives. In the only hook and line (pole-and-line, handline, troll-line and longline) gears are used for fishing. Other gears such as seines, gillnet and troll nets are not utilised. The target species is mainly skipjack and yellowfin tuna. 

We had added new information on the PCA and discussed some of the hypothesis related to tuna aggregations. 

Responses to comments made on the pdf document.

Comment 1: 

Line #45: Although this was talking about the fishery in general, somehow it lacks a general introduction on different fishing vessels and their characteristics that utilize the AFADs. How about hook and lines, if they go and fish near these AFADs are they not allowed? Government regulation that only pole and lines are allowed to fish on AFADs have not also been mentioned.

Response:

We have included additional information on the pole-and-line fishing vessels and the regulation related to fishing at the AFADs.

“fishers moved from small wooden sail boats to large (25 to 30m in length) mechanized vessels [2]. Throughout the Maldives only hook and line (pole-and-line, handline, trolling and longline) fishing is practised. Although longline fishing for tuna was carried, it has been suspended since July 2019 in the Maldives. Other gears such as purse seine, gill nets or trawl nets are forbidden to ensure the sustainability of the stocks. At present there are no foreign fishing vessels operating in the Maldives EEZ.”

“Since the government gives priority to the pole-and-line fishery only pole-and-line fishing is permitted within a 3-mile radius of the AFADs.”

Comment 2: 

Line #79: Why are pole-and-line the only fishery allowed? Is this because they do not have other fishing industry or fishing gears in the area. How about the large fishing vessels of other nations …. are they also allowed near its EEZ?

Response:

We have included additional information on fishing gear and operations within the Maldives EEZ.

“Throughout the Maldives only hook and line (pole-and-line, handline, trolling and longline) fishing is practised. Although longline fishing for tuna was carried, it has been suspended since July 2019 in the Maldives. Other gears such as purse seine, gill nets or trawl nets are forbidden to ensure the sustainability of the stocks. At present there are no foreign fishing vessels operating in the Maldives EEZ.”

“About 75% of the tuna catch, mainly skipjack (Katsuwonus pelamis) and small yellowfin (Thunnus albacares) (FL<65cm) tuna, is caught using pole-and-line [3] while the rest is caught by handline and trolling.”

Comment 3: 

Line #84: The large distances between AFADs, is this not related also to the density of pole and lines that visit or fish on the AFADs. 

Response:

The AFADs are evenly distributed across the country and all pole and line fishers have equal access to them. Thus, fishers from extreme south of the country could be seen fishing at an AFAD in the extreme north. 

Comment 4: 

Line #98: Indeed it can be said that it is really long but we are interested to know about their fishing practices which was not mentioned in the introduction. 

Response:

We have added new information on the scale of the fisheries and the gear used. Since the focus is on pole-and-line tuna fishery it is explained more detail.

Comment 5: 

Line #119: The description of the study site is not enough, it lacks details on the fisheries that are based on the north or south of the island …. what specific target species are sought by the fishers … what are the other fisheries that are non-tuna are based in the north or south of the islands. 

Response:

This study focuses on the pole-and-line tuna fishery – especially the tuna related AFAD fishery in the Maldives. The AFAD related tuna fishery across the Maldives target both skipjack and yellowfin tuna using the same technique (pole-and-line) and similar vessels. We feel that it is not within the scope of this paper to include information on other fisheries (coastal fisheries – non tuna fisheries) taking place across the Maldives. Details of the study site is provided with a diagram (figure 1). The study site is the whole of Maldives. Geographical and oceanographic features that may affect the behaviour of tuna are included in the description of the study site. The pole-and-line fishing operation across the Maldives is similar and the target species is the same. As described in the manuscript the catch is higher in the south than in the north. 

“The Maldives extends from about 7oN to 0.5oS stretching 822 km (Fig 1) and is 130 km at its widest. It is subjected to the seasonal monsoons [24, 25] – northeast monsoon is from December to March and the southwest monsoon is from May to October. From the north up to about 2.5oN, the Maldives atolls are arranged in a double chain and below 2.5oN the atolls form a single chain and are separated with wide deep channels through which migratory fish such as tuna travel. In the south of the Maldives, the effect of the monsoon related currents is diminished and influenced by the equatorial systems [24, 25]. More than two thirds of the yearly pole-and-line catch are landed by vessels operated in the south of the Maldives (Average pole-and-line catch for last 5 years: North (7° to 2°N) 16,000t (SD±1000t) and South (2°N to 1°S) 49,500t (SD±6000t). (MoFMRA, 2019)).”

Comment 6: 

Line #120: The fishers that were interviewed, it is unclear how old they were, how long they were fishing in the area and what makes them credible for this type of work.

Response:

“To ensure the validity of LEK and the quality of information obtained, it is important to select participants who have the appropriate knowledge for the interviews [23]. The fishers for the interviews were selected based on their fishing experience and area of fishing. Hence captains (n=36), deputy captains (n=9) and crew members (n=9) from 36 vessels, who had a minimum of 8 years of experience on a licensed commercial pole-and-line vessel in the Maldives were selected.”

Comment 7: 

Line #183: This should first introduce the characteristics of the fishers or the fishery before directly going to the results of the seasonal variation.

Response:

Characteristics of fishers and fishery were provided in the general introduction. We have added some observations that were made during the interview process. 

Comment 8: 

Line #244: There was a mention of the catch of logbooks but the result was not actually provided or shown here. It would be good to see the results of such collected logbooks and whether it confirms also what the fishers were talking about the fishery. 

Response:

In the results section logbook analysis finding are now included.

“The mid-day positions of pole-and-line fishing vessels obtained from logbooks during the northeast and southwest monsoons (Fig 3) showed a similar trend, with fishers tending to fish more on the east (45% - fishing events) than on the west (32% - fishing events) of the Maldives during the northeast monsoon (Chi-2 test p-value<0.0001) and on the west (42% - fishing events) than on the east (39% - fishing events) of the Maldives during the southwest monsoon (Chi-2 test p-value<0.009). The remaining vessels reported the central region as their mid-day position.”

---

## [Editor Report · Decision Letter 1]

28 May 2021

PONE-D-20-15145R1

Tuna behaviour at anchored FADs inferred from Local Ecological Knowledge (LEK) of pole-and-line tuna fishers in the Maldives

PLOS ONE

Dear Dr. Jauharee,

Thank you for submitting your manuscript to PLOS ONE. After careful consideration, we feel that it has merit but does not fully meet PLOS ONE’s publication criteria as it currently stands. Therefore, we invite you to submit a revised version of the manuscript that addresses the points raised during the review process.

The authors have addressed all the reviewer comments.  There are only four minor editorial issues:

* ABSTRACT

- Line 24:  Fix the verb tense to ensure they are consistent within the sentence:

"The Maldives tuna fishery landings in 2018 WERE 148, 000 t and accountED for nearly a 25 quarter of the global pole-and-line tuna catch." 

- Line 33:  Rewrite this sentence: "abundant on the eastern SIDE OF THE MALDIVES, while during the southwest monsoon they are more abundant on the western side."  

Or alternatively:  "during the northeast and southwest monsoons, tuna are more abundant on the eastern and western sides of the Maldives, respectively"

* INTRODUCTION:

- Please rephrase this sentence:

"Fish behaviour is a key element of scientific expertise to assist in stock assessment and fisheries management "

I would suggest: "Understanding fish behaviour is key for stock assessment and fisheries management "  

* METHODS AND RESULTS:

Please report the df and the chi-square test statistic, when reporting the results oF the chi-square tests:

"Contingency test, Chi-square = X, df = N,  p-value<0.009"

We look forward to receiving your revised manuscript.

Kind regards,

David Hyrenbach, Ph.D.

Academic Editor

PLOS ONE
---

## [Author Response · Author response to Decision Letter 1]

29 Jun 2021

Dear David Hyrenbach, Academic Editor,

We have addressed the comments made in the decision letter and attached are the figures processed by PACE. 

Thank you very much. We really appreciate all the support provided by PLOS editorial.

Best regards

Riyaz Jauharee

---

## [Editor Report · Decision Letter 2]

1 Jul 2021

Tuna behaviour at anchored FADs inferred from Local Ecological Knowledge (LEK) of pole-and-line tuna fishers in the Maldives

PONE-D-20-15145R2

Dear Dr. Jauharee,

Your second revision successfully addressed all the editorial suggestions of the editor.  Thus, we are pleased to inform you that your manuscript has been judged scientifically suitable for publication and will be formally accepted for publication once it meets all outstanding technical requirements.  

Kind regards,

David Hyrenbach, Ph.D.

Academic Editor

PLOS ONE
---

## [Editor Report · Acceptance letter]

19 Jul 2021

PONE-D-20-15145R2 

Tuna behaviour at anchored FADs inferred from Local Ecological Knowledge (LEK) of pole-and-line tuna fishers in the Maldives 

Dear Dr. Jauharee:

I'm pleased to inform you that your manuscript has been deemed suitable for publication in PLOS ONE. Congratulations! Your manuscript is now with our production department. 

Kind regards, 

on behalf of

Dr. David Hyrenbach 

Academic Editor

PLOS ONE